# Large Multilingual Models Pivot Zero-Shot Multimodal Learning across Languages

**Jinyi Hu**[1]    **Yuan Yao**[1*]   **Chongyi Wang**[5]    **Shan Wang**[5]   **Yinxu Pan**[6]
**Qianyu Chen**[1]   **Tianyu Yu**[1]   **Hanghao Wu**[6]   **Yue Zhao**[2]   **Haoye Zhang**[1]
**Xu Han**[1,3]    **Yankai Lin**[4]   **Jiao Xue**[5]   **Dahai Li**[5]   **Zhiyuan Liu**[1*]    **Maosong Sun**[1*]
[1] Tsinghua University    [2] Beijing University of Posts and Telecommunications
[3] Shanghai Artificial Intelligence Laboratory    [4] Renmin University of China
[5] Zhihu Inc.    [6] ModelBest Inc.
`hu-jy21@mails.tsinghua.edu.cn`

## Abstract

Recently there has been a significant surge in multimodal learning in terms of both image-to-text and text-to-image generation. However, the success is typically limited to English, leaving other languages largely behind. Building a competitive counterpart in other languages is highly challenging due to the low-resource nature of non-English multimodal data (i.e., lack of large-scale, high-quality image-text data). In this work, we propose MPM, an effective training paradigm for training large multimodal models in non-English languages. MPM demonstrates that **M**ultilingual language models can **P**ivot zero-shot **M**ultimodal learning across languages. Specifically, based on a strong multilingual large language model, multimodal models pretrained on English-only image-text data can well generalize to other languages in a (quasi)-zero-shot manner, even surpassing models trained on image-text data in native languages. Taking Chinese as a practice of MPM, we build large multimodal models VisCPM in image-to-text and text-to-image generation, which achieve state-of-the-art (open-source) performance in Chinese. To facilitate future research, we open-source codes and model weights at https://github.com/OpenBMB/VisCPM.

## 1 Introduction

With the rapid advancement of powerful models such as GPT-4 (OpenAI, 2023) and Stable Diffusion (Rombach et al., 2022) in their multimodal capabilities, large multimodal models have emerged as the latest frontier in pursuing achieving Artificial General Intelligence (AGI). Generally, the multimodal generative capabilities across images and text can be divided into two categories: (i) In the field of image-to-text generation, prominent multimodal large language models like GPT-4 (OpenAI, 2023), LLaVA (Liu et al., 2023a) and InstructBLIP (Dai et al., 2023) exhibit remarkable multimodal conversational and reasoning abilities based on images; (ii) In the field of text-to-image generation, models such as Imagen (Saharia et al., 2022) and Stable Diffusion (Rombach et al., 2022) excel in generating highly realistic and relevant images based on text prompts. These models possess exceptional capabilities in processing images and text, profoundly reshaping the landscape of multimodal AI in both academia and industry.

However, the success of large multimodal models has mainly been achieved within the English community, while the multimodal capabilities in other non-English languages significantly trail behind. Bridging this gap is challenging due to the extensive image-text pair data requirements for training multimodal models. For instance, the pretraining of BLIP-2 (Li et al., 2023a) involves more than 100M high-quality image-text pairs, while Stable Diffusion (Rombach et al., 2022) utilizes more than 2B pairs. As a result of the paucity of such multimodal data resources in non-English languages, the progress of multimodal research in these languages remains hindered.

To address this challenge, we propose MPM, an effective training paradigm for large multimodal models in non-English languages. MPM utilizes the **M**ultilingual language model to **P**ivot **M**ultimodal

---

*Corresponding authors.

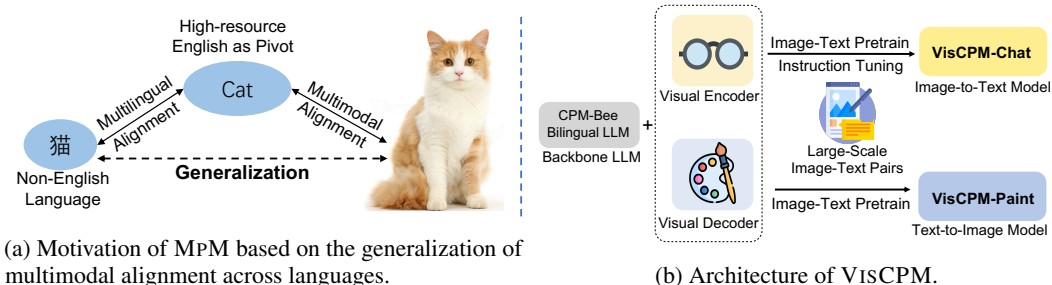

(a) Motivation of MPM based on the generalization of multimodal alignment across languages.

(b) Architecture of VISCPM.

Figure 1: Overview of the motivation and architecture of MPM and VISCPM.

learning across languages and considers English, which contains substantial multimodal data resources, as a pivot between the visual signals and non-English languages which commonly lacking in multimodal data. MPM draws inspiration from the *Bilingual Dual-coding Theory* (Paivio, 2014; Clark & Paivio, 1991) that argue that *visual semantics are largely language-agnostic*. Intuitively, as portrayed in Fig. 1a, multilingual learners can effectively align the visual semantics with newly acquired language based on established multimodal and multilingual alignment. Simulating the human learning process, MPM also divides the non-English multimodal learning into two consecutive stages: multilingual alignment and multimodal alignment. The former focuses on building a multilingual model, while the latter culminates in a multimodal model spanning multiple languages.

Specifically, for multilingual alignment, MPM harnesses a pretrained multilingual large language model (LLM) as the backbone language model, which can provide aligned representations for different languages. Next, for the multimodal alignment, MPM trains the visual modules based on the multilingual model exclusively on English image-text pairs to align English and visual semantics. Similar to how humans learn, using the multilingual model as a pivot point, the resultant multimodal model naturally acquires zero-shot multimodal capability in other non-English languages.

Taking Chinese as a practical instance for MPM, we develop Chinese large multimodal models named VISCPM based on English-Chinese bilingual large language model CPM-Bee (Zhang et al., 2021). Notably, pretraining exclusively on English image-text pairs, the zero-shot performance of VISCPM in Chinese still surpasses that of existing Chinese multimodal models trained on image-text pairs in native Chinese. The promising performance of MPM in Chinese sheds light on its potential application in broader languages. Following the same training process, we further extend MPM to develop a multilingual multimodal conversation model supporting six languages based on the LLaMA (Touvron et al., 2023), including English, German, French, Spanish, Italian, and Portuguese.

In summary, the contributions of this paper are as follows: (i) We propose MPM, an effective training paradigm specifically designed for non-English languages lacking multimodal resources. Researchers worldwide can utilize MPM to rapidly adapt advanced multimodal learning methods from English to their respective languages. (ii) We develop a series of Chinese large multimodal models VISCPM as a practical application of MPM, which achieves state-of-the-art performance among open-source Chinese multimodal models. (iii) We open-source the model weights of VISCPM and provide experimental details, serving as a valuable reference for other researchers. (iv) We validate the generalization capability of VISCPM in diverse languages and develop the multilingual multimodal conversation model across six languages.

## 2 RELATED WORK

**Image-to-text Models.** Traditional image-to-text generation models mainly focus on the task of image caption and visual question answering (Yuan et al., 2021; Yu et al., 2022a; Wang et al., 2022b). Recently, the mainstream of image-to-text has turned to multimodal LLM, focusing on rendering LLM capable of multimodal interaction with users. These models connect the visual module and LLM with perceivers, such as BLIP-2 (Li et al., 2023a) and InstructBLIP (Dai et al., 2023) or linear projectors such as LLaVA (Liu et al., 2023a) and PandaGPT (Su et al., 2023). VPGTrans (Zhang et al., 2023) explores the transferability of visual modules across LLM. Meanwhile, many efforts have been dedicated to building multimodal instruction following datasets. LLaVA (Liu et al., 2023a)

and MiniGPT-4 (Zhu et al., 2023) build visual content-related dialog by transferring image captions into conversation data using GPT-4. InstructBLIP (Dai et al., 2023) and M³IT (Li et al., 2023b) incorporate downstream vision-language datasets to construct instruction data. LLaVA-RLHF (Sun et al., 2023) and RLHF-v (Yu et al., 2023b) propose multimodal RLHF for trustworthy behavior.

**Text-to-image Models.** In the early stages of text-to-image model development, generative adversarial networks (Zhu et al., 2019; Li et al., 2019) and auto-regressive generation (Esser et al., 2021) are popularly chosen architectures for text-to-image synthesis models (Li et al., 2019). More recently, large-scale diffusion-based text-to-image models such as DALLE-2 (Ramesh et al., 2022), Imagen (Saharia et al., 2022), and Stable Diffusion (Rombach et al., 2022) have taken center stage, demonstrating exceptional generative capabilities.

**Multilingual Multimodal Models.** The extension of multimodal models to include more languages has become a key research focus over the past few years. Researchers have made efforts to extend the powerful image-text model CLIP (Radford et al., 2021) to handle more languages using techniques of knowledge distillation (Carlsson et al., 2022; Hu et al., 2023) or contrastive learning (Bianchi et al., 2021; Chen et al., 2023). Other studies have aimed to create a universal framework for multilingual vision-language pertaining and simultaneously achieve multilingual and multimodel alignment (Ni et al., 2021; Zhou et al., 2021; Zeng et al., 2023). In the era of LLMs, PaLI (Chen et al., 2022) develops a 17B multilingual language-image model based on 10B image-text pairs spanning 100 languages. Ying-VLM (Li et al., 2023b) shows that instruction tuning in English can generalize to other languages. MultiFusion (Bellagente et al., 2023) discover that the multilingual language model can help cross-lingual transfer in text-to-image generation. In comparison, this work provides a more systematical formulation for training multilingual multimodal models and demonstrates that the zero-shot transfer performance of these models can surpass that of models trained on native-language multimodal data.

## 3 MPM TRAINING PARADIGM

In this section, we first present the formulation of multilingual multimodal learning and provide an overview of the training procedure of MPM. Following this, we detail the specific training procedures of MPM for both image-to-text and text-to-image generation.

### 3.1 PROBLEM FORMULATION AND OVERVIEW

Multimodal learning can be formulated as modeling the relationship between images, denoted as $x$, and text, denoted as $y$, in a target language $l_t$. In this context, the image-to-text generation, which can be roughly summarized as generating description for input images, is to learn the conditional distribution $p_\theta(y^{l_t}|x)$ parameterized by $\theta$; the text-to-image generation, which is to synthesize relevant images given input text prompts, is to learn $p_\phi(x|y^{l_t})$ parameterized by $\phi$.

In the vanilla setting, these conditional distributions are typically trained using image-text pairs $\mathcal{D}_t = \{(x_i, y_i^{l_t})\}_{i=1}^N$ in the target language $l_t$ (Radford et al., 2021; Yang et al., 2022). However, high-quality image-text pairs are extremely scarce in most languages. To mitigate the dependency on native image-text pairs, we introduce the pivot language $l_p$, which contains abundant multimodal pairs $\mathcal{D}_p = \{(x_i, y_i^{l_p})\}_{i=1}^M$, where $M \gg N$. Imitating the human learning mechanism that can naturally align visual concepts with various learned languages, MPM aims to transfer visual concepts learned in the pivot language to the target language.

MPM divides the multimodal learning process in target language $l_t$ into two consecutive stages: **multilingual alignment** and **multimodal alignment**. For the multilingual alignment, MPM aims to establish the cross-lingual alignment for $l_t$ and $l_p$. This is achieved by directly leveraging a pretrained multilingual LLM, denoted as $f_\sigma$, which can provide close hidden representations for text pair $y^{l_t}$ and $y^{l_p}$ with similar semantics, i.e., $f_\sigma(y^{l_t}) \approx f_\sigma(y^{l_p})$. For the multimodal alignment, MPM utilize the sufficient multimodal resource $\mathcal{D}_p$ in the pivot language and optimize the image-to-text objective $p_\theta(y^{l_p}|x)$ and text-to-image objective $p_\phi(x|y^{l_p})$. In the following sections, we introduce the training process of **multimodal alignment** stage. It's worth noting that MPM is agnostic to the specific model architecture and training method, which enables us to flexibly utilize existing highly effective model architectures and training techniques in each task.

## 3.2 IMAGE-TO-TEXT GENERATION

In image-to-text generation, we incorporate an image encoder module $h_\xi$ parameterized by $\xi$ to provide visual feature $\mathbf{z} = h_\xi(x)$. These visual features $\mathbf{z}$ are then concatenated with the text embedding as input into the multilingual LLM. Following recent work to train multimodal conversation models (Zhu et al., 2023; Liu et al., 2023a), MPM's training process for image-to-text generation consists of two sub-stages: Multimodal Pretraining and Instruction Tuning.

**Multimodal Pretraining.** In this sub-stage, we pretrain the visual module to align it with LLM on a large scale of image-text pairs using the language modeling objective:

$$\mathcal{L}_1(p_\theta, \mathcal{D}_p) = -\sum_{i=1}^{M} \log p_\theta(y_i^{l_p} | h_\xi(x_i)). \tag{1}$$

Here, we fix the parameters of LLM ($\theta = \{\xi\}$) to prevent the powerful capabilities of LLM from being influenced by short texts in the image-text pairs.

**Instruction Tuning.** To enhance models' capabilities in following human instructions, we conduct instruction tuning on elaborately curated multimodal instruction tuning datasets built by blending the existing multimodal instruction tuning datasets in the pivot language and their translated version in the target language. We denote this multilingual instruction tuning datasets as $\mathcal{D}_i = \{x_k, y_{q,k}^l, y_{a,k}^l\}_{k=1}^S$, where $y_q^l$ is the instructions and $y_a^l$ is the response in certain language $l$. Both the visual module and multilingual LLM are fine-tuned, i.e., $\theta = \{\xi, \sigma\}$, by maximizing the probability of the response:

$$\mathcal{L}_2(p_\theta, \mathcal{D}_i) = -\sum_{k=1}^{S} \log p_\theta(y_{a,k}^l | h_\xi(x_k), f_\sigma(y_{q,k}^l)). \tag{2}$$

Interestingly, we find a *quasi-zero-shot* transfer capability of multilingual multimodal models in this scenario. If excluding the translated variant in the target language and solely performing instruction tuning using the pivot language, when given an image $x$ and a question or an instruction $y_q^{l_t}$ in the target language, the resultant model responds accurately though mostly in the pivot language. This can be attributed to the close resemblance between the hidden representation of instructions in two languages provided by the multilingual LLM, i.e., $f_\sigma(y_q^{l_p}) \approx f_\sigma(y_q^{l_t})$. Consequently, we have $p_\theta(y_a^{l_p} | h_\xi(x), f_\sigma(y_q^{l_p})) \approx p_\theta(y_a^{l_p} | h_\xi(x), f_\sigma(y_q^{l_t}))$. Since both the pretraining and instruction tuning stages employ text components solely in the pivot language, the LLM can understand the question in the target language but cannot calibrate the response in the same language.

To stimulate the model to respond in the target language, MPM incorporates a small number of translated pairs in the target language during instruction tuning. In this way, MPM simultaneously improves the model's instruction-following capability and calibrates the response language, ultimately realizing a multimodal chatbot in the target language.

## 3.3 TEXT-TO-IMAGE GENERATION

In the text-to-image generation, we adopt a similar architecture with Stable Diffusion (Rombach et al., 2022). It incorporates a denoising network $g_\delta$ with a UNet architecture (Ronneberger et al., 2015) parameterized by $\delta$ as an image decoder to generate images given the input prompt. The LLM $f_\sigma$ and image decoder $g_\delta$ are interconnected with cross-attention mechanism (Vaswani et al., 2017).

Diffusion models (Ho et al., 2020; Song et al., 2020) involve learning an iterative process of denoising a Gaussian noise into data distribution. The denoise network is optimized to remove the noise of noised image $x_\tau$, conditioned on the hidden states of text input provided by the LLM. The training objective is defined as follows:

$$\mathcal{L}(p_\phi, \mathcal{D}_p) = \mathbb{E}_{x, y^{l_p}, \varepsilon, \tau}[||g_\delta(x_t, f_\sigma(y^{l_p}), \tau) - \varepsilon||_2^2], \tag{3}$$

In this stage, $\phi = \{\delta\}$, i.e., the image decoder is trained to align with frozen LLM. In this way, when input with the unseen prompt in the target language $y^{l_t}$, the multilingual LLM $f_\sigma$ can inherently provide a representation $f_\sigma(y^{l_t})$ close to the seen representation $f_\sigma(y^{l_p})$ of the pivot language prompt with similar semantics. Therefore, the capability of text-to-image in the target language can be seamlessly transferred from the pivot language in a zero-shot fashion, illustrated as $g_\delta(x_\tau, f_\sigma(y^{l_t}), \tau) \approx g_\delta(x_\tau, f_\sigma(y^{l_p}), \tau)$.

## 4 VisCPM

As a practice of MPM, we develop a series of large-scale Chinese multimodal models called VisCPM. We use Chinese as the target language and English as the pivot language. The Chinese-English bilingual language model CPM-Bee (Zhang et al., 2021) serves as the backbone multilingual LLM. We have two variations of the model: VisCPM-Chat for image-to-text multimodal conversation and VisCPM-Paint for text-to-image synthesis. In the following sections, we begin by providing an overview of existing multimodal datasets in Chinese and then introduce the training procedure of VisCPM-Chat and VisCPM-Paint.

### 4.1 Are Chinese Multimodal Datasets Enough To Train A Multimodal Model?

The largest publicly available multimodal dataset of native Chinese is Wukong (Gu et al., 2022), consisting of 100M image-text pair pairs. However, by visualizing the CLIP score computed with Chinese-CLIP (Yang et al., 2022), as shown in Fig. 2, and manually inspecting, as introduced in Appendix G, we discover that only a minor fraction of image-text pairs in Wukong possess semantically matched content. The poor quality of the dataset escalates existing data resource shortfalls. A straightforward method to enhance the count of Chinese image-text pairs is to translate the English image-text pairs into Chinese, which has been utilized in previous work (Wang et al., 2022a; Qiu et al., 2022). However, translation requires an external machine translation model, and translating a large-scale dataset used in pretraining consumes substantial computing resources. Also, as discussed in Sec.

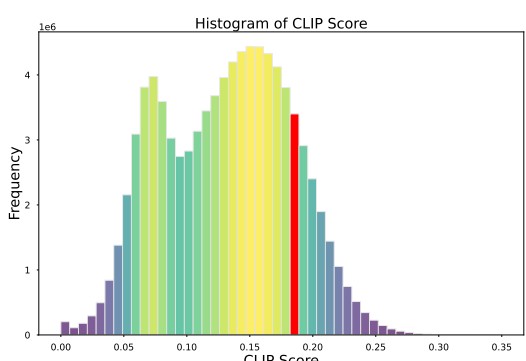

Figure 2: The histogram of CLIP score of 100M Chinese image-text pairs from Wukong dataset. We set 0.18 as a moderate filtering threshold.

5.3, we practically find that incorporating translated image-text only has marginal improvement on the performance when using powerful multimodal LLM as backbone language model, which already possesses strong cross-lingual generalization capability. Based on this analysis, we argue that effectively utilizing the existing English data to achieve knowledge transfer in multimodal alignment is the key to developing a powerful Chinese large multimodal model.

### 4.2 ᘓᘔ VisCPM-Chat

VisCPM-Chat is a Chinese-English bilingual multimodal chatbot capable of responding to users' instructions based on the input image. VisCPM-Chat utilizes the Muffin architecture (Yu et al., 2023a) as the image encoder. Specifically, Muffin directly leverages a pretrained vision-language model BEiT-3 (Wang et al., 2023) as an inherent bridge module between vision and language. In the *multimodal pretraining stage*, the visual module is trained on 100M image-text pairs to align with the frozen LLM for 180K steps. In the *instruction tuning sub-stage*, we utilize bilingual versions of LLaVA 150K (Liu et al., 2023a) and UniMM-Chat (Yu et al., 2023a), and the English part of M³IT (Li et al., 2023b) to fine-tune the image encoder and LLM for 80K steps. The details of these datasets are presented in Appendix B.2. Due to the *quasi-zero-shot* phenomenon in Chinese introduced in Sec. 3.2, we incorporate certain Chinese data by translating LLaVA 150K and UniMM-Chat into Chinese using machine translation.[1]

To demonstrate the effect of Chinese image-text pairs, we also train an additional version of VisCPM-Chat, which adds additional Chinese image-text pairs during pretraining, including 20M native Chinese image-text pairs filtered from 100M Wukong (Gu et al., 2022) and 20M Zero-Chinese (Xie et al., 2022) dataset based on a CLIP score threshold greater than 0.18, and 136M image-text pairs translated from Laion-COCO dataset. We name this model as VisCPM-Chat+.

---

[1] We employ CPM-Bee for machine translation. CPM-Bee can obtain 41.6 BLEU score on WMT 2017 test set (Bojar et al., 2017). Except for special instructions, all translation processes in this work are carried out using the CPM-Bee model.

Table 1: Experimental results on LLaVA Test Set accessed by GPT-4. Con: Conversation, DD: Detailed Description, CR: Complex Reasoning, AVG: the average score of three tasks. The best/second best results are marked in **bold** and underlined, respectively.

| | Model | LLM Backbone | English | | | | Chinese | | | |
|---|---|---|---|---|---|---|---|---|---|---|
| | | | Con | DD | CR | AVG | Con | DD | CR | AVG |
| English Model | MiniGPT-4 (Zhu et al., 2023) | Vicuna-13B | 65.0 | 67.3 | 76.6 | 69.7 | - | - | - | - |
| | InstructBLIP (Dai et al., 2023) | Vicuna-13B | 81.9 | 68.0 | 91.2 | 80.5 | - | - | - | - |
| | LLaVA (Liu et al., 2023a) | Vicuna-13B | **89.5** | **70.4** | 96.2 | **85.6** | - | - | - | - |
| En-Zh Bilingual Model | mPLUG-Owl (Ye et al., 2023) | BLOOMZ-7B | 64.6 | 47.7 | 80.1 | 64.2 | 76.3 | 61.2 | 77.8 | 72.0 |
| | VisualGLM (Du et al., 2022) | ChatGLM-6B | 62.4 | 63.0 | 80.6 | 68.7 | 76.6 | 87.8 | 83.6 | 82.7 |
| | Ziya-Visual (Wang et al., 2022a) | Ziya-LLaMA-13B | 82.7 | 69.9 | 92.1 | 81.7 | 85.0 | 74.7 | 82.4 | 80.8 |
| | Qwen-VL-Chat (Bai et al., 2023) | Qwen-7B | 82.4 | 76.9 | 91.9 | 83.8 | 82.3 | **93.4** | 89.5 | 88.2 |
| | VISCPM-Chat | CPM-Bee-10B | 81.4 | 69.2 | 93.1 | 81.4 | 90.0 | 87.4 | 95.0 | 90.9 |
| | VISCPM-Chat+ | CPM-Bee-10B | 80.1 | 67.1 | **97.1** | 81.5 | **91.3** | 90.7 | **95.4** | **92.5** |

## 4.3 🎨 VISCPM-PAINT

VISCPM-Paint is a text-to-image synthesis model that can accept prompts in both Chinese and English. VISCPM-Paint employs the UNet in Stable Diffusion (Rombach et al., 2022) as the image decoder. To maintain the generative capability of UNet, the training process only involves the cross-attention layer of UNet and the linear transblock between the multilingual LLM and UNet. We optimize these parameters using an extensive dataset of English image-text pairs Laion-2B (Schuhmann et al., 2022) for 300K steps.

Similar to VISCPM-Chat+, we train an additional version of VISCPM-Paint, which is fine-tuned on Chinese image-text pairs. The component of these pairs are identical to that VISCPM-Chat+ uses. We name this model as VISCPM-Paint+.

## 5 EXPERIMENTS

### 5.1 EVALUATION OF VISCPM-CHAT

#### 5.1.1 EVALUATION SETTING

**Baselines**. We compare VISCPM-Chat with existing multimodal conversation models, which include the English-only models: MiniGPT-4 (Zhu et al., 2023), InstructBLIP (Dai et al., 2023), and LLaVA (Liu et al., 2023a), as well as Chinese-English bilingual models: mPLUG-Owl (Ye et al., 2023), VisualGLM (Du et al., 2022), Ziya-Visual (Wang et al., 2022a), and Qwen-VL-Chat (Bai et al., 2023). All of these Chinese-English models have performed large-scale training on both Chinese and English multimodal datasets. More details of baselines are presented in Appendix C.

**Evaluation Benchmark**. We assess the multimodal conversational abilities of the VISCPM-Chat in both English and Chinese on the LLaVA Test Set (Liu et al., 2023a) and UniMM-Bench (Yu et al., 2023a). We use CPM-Bee to translate them into Chinese and manually check the translation results. Specifically, LLaVA comprehensively evaluates the model's multimodal conversation capability from "conversation", "detailed description", and "complex reasoning". UniMM-Bench, drawn from commonly used visual question-answering datasets—including OKVQA (Marino et al., 2019), AOKVQA (Schwenk et al., 2022), GQA (Hudson & Manning, 2019), and VQAv2 (Antol et al., 2015)—is designed particularly for evaluating multimodal conversation models on the abilities associated with reasoning, commonsense, and world knowledge. Considering that multimodal conversation models commonly generate responses with complete sentences to answer the question, traditional string-matching-based metrics, such as VQAScore (Antol et al., 2015), are not suitable in this context. Therefore, for the LLaVA Test Set and UniMM-Bench, we follow Liu et al. (2023a); Yu et al. (2023a) and employ GPT-4[2] to rate the model-generated responses and the reference answers. Further details of the LLaVA Test Set and UniMM-Bench are presented in Appendix D.3.

---

[2]Specifically, we use the GPT-4-0314 version to evaluate the responses.

Table 2: Evaluation of bilingual multimodal models on UniMM-Bench in Chinese. See results in English at Table 6.

| Model | OKVQA | AOKVQA | GQA | VQAv2 | AVG |
|---|---|---|---|---|---|
| mPLUG-Owl | 52.8 | 55.8 | 60.8 | 56.7 | 56.5 |
| VisualGLM | 53.7 | 57.5 | 64.8 | 62.5 | 59.6 |
| Ziya-Visual | 59.2 | 58.1 | 61.9 | 59.1 | 59.6 |
| Qwen-VL-Chat | 58.5 | 58.5 | **70.6** | **72.0** | 64.9 |
| VisCPM-Chat | **62.8** | **64.9** | 68.3 | 71.8 | **67.0** |

Table 3: Results of extension to more languages based on Vicuna.

| Lang | Con | DD | CR | AVG |
|---|---|---|---|---|
| *en* | 87.6 | 76.8 | 101.0 | 88.6 |
| *de* | 90.8 | 80.8 | 93.6 | 88.7 |
| *fr* | 87.6 | 81.9 | 94.6 | 88.2 |
| *es* | 87.1 | 79.6 | 102.3 | 89.8 |
| *it* | 84.5 | 79.5 | 93.3 | 85.9 |
| *pt* | 81.3 | 81.6 | 92.4 | 85.1 |

### 5.1.2 EXPERIMENTAL RESULTS

**Quantitative Results.** The evaluation results on the LLaVA Test Set and UniMM-Bench are presented in Table 1 and Table 2. In **Chinese**, VISCPM-Chat achieves remarkable results and outperforms all baseline models on both benchmarks. It is worth noting that, unlike VisualGLM and Qwen-VL-Chat, which leverage a substantial number of Chinese image-text pairs during pretraining, VISCPM-Chat does not incorporate any Chinese multimodal data in its pretraining process. Despite this, VISCPM-Chat still presents superior performance on conversational abilities, commonsense, and world knowledge. Such results strongly demonstrate the effectiveness of MPM in transferring visual knowledge from English to Chinese. In **English**, the performance of VISCPM-Chat exceeds that of mPLUG-Owl, VisualGLM, InstructBLIP, and MiniGPT-4, and is roughly on par with Ziya-Visual and Qwen-VL-Chat while remaining comparable to the strong baseline LLaVA.

**Case Study.** In addition to acquiring broad world knowledge and commonsense, MPM training also implicitly stimulated VISCPM-Chat's competency in transferring knowledge unique to Chinese culture. For example, as shown in Fig. 3, For instance, as illustrated in Fig. 3, VISCPM-Chat is capable of relating the Chinese classical poems present in Su Shi's "Water Melody" according to the full moon and pavilion. Although this image-text pair deeply related to Chinese culture is not included in the training data, VISCPM-Chat successfully navigates such a non-trivial knowledge transfer through MPM training. For a more detailed analysis of cases involving VISCPM-Chat, please refer to Appendix B.3.

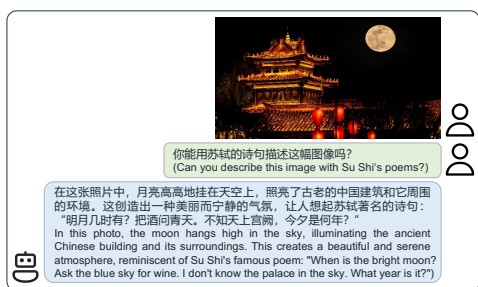

Figure 3: Multimodal conversation cases of VISCPM-Chat in Chinese.

### 5.1.3 GENERALIZATION TO MORE LANGUAGES

The remarkable results of VISCPM-Chat on Chinese of MPM illuminate the potential broader application to a more diverse set of languages. Specifically, we leverage the multilingual LLM LLaMA (Touvron et al., 2023) as the LLM backbone and consider German, French, Spanish, Portuguese, and Italian as target languages. Following the same training procedure as in VISCPM-Chat, we develop a multilingual multimodal chatbot proficiently supporting six languages. We begin by pretraining the visual encoder with LLaMA to achieve visual feature alignment in English image-text pairs. Next, we employ M2M-100 (Fan et al., 2021) to translate the instruction training set of LLaVA into five target languages. The original English instruction training set and the five translated sets are merged and shuffled, then used for fine-tuning the visual module and LLM.

Table 3 presents the evaluation results for English and five target languages on LLaVA Testset. See the results on IGLUE benchmark (Bugliarello et al., 2022) in Appendix D.5. Notably, for the relatively popular languages, such as German, French, and Spanish, the average scores exceed 88. Additionally, for Italian and Portuguese, the average scores are above 85. These results are highly encouraging as they demonstrate the chatbot's coherent and accurate responses to visual-related questions in all six languages, even though the five target languages are simply blended during instruction tuning. The results in these languages validate the generalization and robustness of MPM in building powerful multimodal models in diverse linguistic contexts.

Table 4: Zero-shot FID on MSCOCO dataset.

| Model | FID↓ | |
|---|---|---|
| | En | Ch |
| GLIDE (Nichol et al., 2022) | 12.2 | - |
| Make-A-Scene (Gafni et al., 2022) | 11.8 | - |
| DALL·E-2 (Ramesh et al., 2022) | 10.4 | - |
| UniDiffuser (Bao et al., 2023) | 9.7 | - |
| CogView2 (Ding et al., 2022) | - | 24.0 |
| Stable Diffusion (Rombach et al., 2022) | **8.6** | - |
| AltDiffusion (Chen et al., 2023) | 17.2 | 16.1 |
| TaiyiDiffusion (Wang et al., 2022a) | - | 15.6 |
| VISCPM-Paint | 9.5 | 10.9 |
| VISCPM-Paint+ | 9.9 | **9.6** |

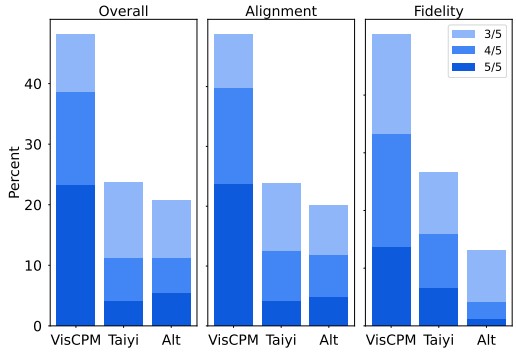

Figure 4: Results of human evaluation for text-to-images on Chinese Drawbench.

## 5.2 EVALUATION OF VISCPM-PAINT

### 5.2.1 EVALUATION SETTING

We compare VISCPM-Paint with several strong text-to-image model, which including English-only models: GLIDE (Nichol et al., 2022), Make-A-Scene (Gafni et al., 2022), DALL·E-2 (Ramesh et al., 2022), UniDiffuser (Bao et al., 2023), and Chinese or Chinese-English bilingual text-to-image models: CogView2 (Ding et al., 2022), AltDiffusion (Chen et al., 2023) and TaiyiDiffusion (Wang et al., 2022a). We mainly compare VISCPM-Paint with AltDiffusion (Chen et al., 2023) and TaiyiDiffusion (Wang et al., 2022a). More details of baselines are presented in Appendix C.

### 5.2.2 AUTOMATIC EVALUATION

For the text-to-image tasks, we assess the zero-shot Frechet Inception Distance (FID) (Heusel et al., 2017) and the CLIP score (Ramesh et al., 2021) using the MSCOCO validation set (Lin et al., 2014). We sample 30K prompts from MSCOCO, and for the Chinese evaluations, the text prompts are translated from the original English captions. We maintain the same sampling configuration for VISCPM-Paint, AltDiffusion, and TaiyiDiffusion and grid search the optimal FID across eight separate classifier guidance scales.

We present the zero-shot FID on MSCOCO validation in Table 4. In **Chinese**, VISCPM-Paint achieves the best FID performance. By solely training on English image-text pairs, VISCPM-Paint displays a significant advantage over AltDiffusion (Chen et al., 2023) and TaiyiDiffusion (Wang et al., 2022a). In **English**, the performance of VISCPM-Paint is comparable to existing powerful text-to-image models, such as Stable Diffuson (Rombach et al., 2022) and UniDiffuser (Bao et al., 2023). See more analysis of fidelity and alignment trade-offs in Appendix E.2.

### 5.2.3 HUMAN EVALUATION

Following previous work (Yu et al., 2022b; Chang et al., 2023), we perform a human evaluation to have a more comprehensive understanding of model performance. Due to the lack of human evaluation benchmarks in Chinese, we create a Chinese text-to-image human evaluation benchmark, named *Chinese Drawbench*. *Chinese Drawbench* consists of 174 prompts and evaluates text-to-image models' proficiency across different aspects. We conduct a human evaluation involving VISCPM-Paint, AltDiffusion (Chen et al., 2023), and TaiyiDiffusion (Wang et al., 2022a) on *Chinese Drawbench*. We ask five independent annotators to judge the best image generated by three models for each prompt. Their evaluation criteria included Overall, Alignment, and Fidelity. More details of human evaluation and *Chinese Drawbench* are provided in Appendix E.3.

Figure 4 presents the human evaluation results. The figure shows the preference shares of 5 evaluators for each generated image with different marginals over consensuses in three aspects. Notably, VISCPM-Paint receives the strongest preference across all three evaluation aspects. In each aspect, more than 40 percent of images generated by VISCPM-Paint are the favored choice. Impressively, in the more objective metrics of Overall and Alignment, VISCPM-Paint earns more than 20 percent 5/5

Table 5: Performance in Chinese with different combinations of dataset languages in each training stage. IT means instruction tuning.

(a) Image-to-text: Evaluation results on LLaVA Test Set.

| Dataset Language | | Con | DD | CR | AVG |
|---|---|---|---|---|---|
| Pretrain | IT | | | | |
| Native Chinese | Chinese | 82.4 | 81.8 | 91.7 | 85.5 |
| English | Chinese | 84.0 | **83.6** | 91.2 | 86.3 |
| English | Bilingual | 85.4 | 81.4 | **96.6** | 88.0 |
| English+ Native Chinese | Bilingual | 89.1 | 82.3 | 91.4 | 87.8 |
| English+ Native Chinese+ Translated Chinese | Bilingual | **90.3** | 81.4 | 92.1 | **88.2** |

(b) Text-to-image: zero-shot FID on MSCOCO.

| Dataset Language | | FID↓ |
|---|---|---|
| Pretrain | Finetune | |
| Native Chinese | - | 15.1 |
| English | - | 10.9 |
| English | Native Chinese w/o Filter | 12.8 |
| English | Native Chinese | 10.7 |
| English | Native Chinese + Translated Chinese | **9.6** |

preference. These results strongly demonstrate the superior quality of VISCPM-Paint's generated images compared to the two baseline models.

## 5.3 ABLATION STUDY

We conduct the ablation study of dataset languages to investigate the impact of varied dataset language combinations on multimodal models' performance in image-to-text and text-to-image tasks. For better efficiency, we only use LLaVA 150K for instruction tuning in the image-to-text task. The detailed configurations of each experiment are reported in Appendix F. Based on the results presented in Table 5a and Table 5b, We have the following observations: (i) *Relying solely on a native Chinese dataset is insufficient for achieving good performance in both image-to-text and text-to-image tasks.* Models trained exclusively on native Chinese datasets yield worse scores, with the image-to-text obtaining an 85.5 average score and the text-to-image model attaining a 15.1 FID. (ii) *English data plays a crucial role in improving the chat capability of the model in Chinese during the instruction tuning stage.* When the image-to-text model is pretrained on a large dataset of English but then fine-tuned using the monolingual Chinese instruction tuning dataset, its average performance experiences a decline from 88.0 to 86.3 compared to the model utilizing the bilingual instruction tuning dataset. (iii) *The filtering process applied to the native dataset is essential for the Chinese performance.* In the text-to-image task, after pretraining on English data and then fine-tuning with unfiltered native Chinese multimodal pairs, the FID worsens from 10.9 in the zero-shot scenario to 12.8. (iv) *Incorporating the native Chinese dataset and translated Chinese dataset yields a marginal improvement in the model performance.* Specifically, adding native Chinese data into the pretraining stage of the image-to-text model or fine-tuning stage of the text-to-image model only results in a 0.2 variation of metrics, while further mixing translated datasets improves the FID from 10.7 to 9.6. The improvement from VISCPM-Chat to VISCPM-Chat+, shown in Table 1, also confirms the same results. We present a more systematic exploration of the influence on the response language accuracy and content quality with different percentages of the Chinese instruction tuning datasets in Appendix D.4.

## 6 CONCLUSION

In this work, we introduce MPM, an innovative training paradigm designed for effectively training large multimodal models in non-English languages. By utilizing a multilingual LLM as a pivotal intermediary between vision signals and target languages, MPM facilitates the efficient transfer of multimodal alignment knowledge across different languages. Based on MPM, we develop a series of open-sourced Chinese large multimodal models VISCPM, which show remarkable capability in Chinese image-to-text and text-to-image tasks. Our experimental results demonstrate that by solely relying on English multimodal data, VISCPM can achieve the SOTA performance among Chinese open-sourced multimodal models. We further scale the scope of language by constructing a versatile multimodal chatbot that supports six distinct languages. We believe that the effectiveness of MPM will contribute to the development of large multimodal models worldwide, thereby fostering the development of sophisticated multimodal models across various languages and cultures.

ACKNOWLEDGEMENT

This work is supported by the National Key R&D Program of China (No.2022ZD0160501).

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

APPENDIX

## A  CONTRIBUTIONS

The authors' contributions can be outlined as follows:

In the preparation of the project, Jinyi Hu and Yuan Yao design the model architecture. Xu Han, Yankai Lin, Jiao Xue, Dahai Li, Zhiyuan Liu, and Maosong Sun offer invaluable guidance in refining the model's architecture. Shan Wang, Chongyi Wang, and Jinyi Hu take charge of collecting and processing the extensive multimodal dataset required for pretraining. Additionally, Hanghao Wu, Yue Zhao, Haoye Zhang, and Yuan Yao collaborate on constructing the instruction tuning data. Jinyi Hu, Chongyi Wang, Tianyu Yu, Qianyu Chen, and Shan Wang jointly implement the training codebase.

In model training, Chongyi Wang, Tianyu Yu, and Yinxu Pan babysit the VISCPM-Chat training; Jinyi Hu, Shan Wang and Qianyu Chen take care of the VISCPM-Paint training.

In model evaluation, Jinyi Hu and Yuan Yao design the evaluation framework. Chongyi Wang and Tianyu Yu evaluate VISCPM-Chat; Jinyi Hu and Shan Wang execute the automatic evaluation of VISCPM-Paint and organize the human evaluation of VISCPM-Paint.

In paper writing, Jinyi Hu and Yuan Yao write the main paper; Yankai Lin, Zhiyuan Liu, and Maosong Sun provide suggestions to polish the writing.

For public usability, Jinyi Hu, Yinxu Pan, Chongyi Wang, Shan Wang, and Yuan Yao promote the open-source of VISCPM; Yinxu Pan develops the online demo and API of VISCPM; Chongyi Wang and Yinxu Pan implement the low-resource inference of VISCPM.

Throughout the project, Xu Han, Yankai Lin, Jiao Xue, Dahai Li, Zhiyuan Liu, and Maosong Sun provide invaluable technical guidance and advice.

## B  DATASET

### B.1  PRETRAINING DATASET

**COCO** (Lin et al., 2014): The COCO dataset is a meticulously compiled image caption dataset that encompasses everyday scenes with common objects. It includes a training set of 118,287 images, each complemented by five unique captions. For each pass through the dataset, we randomly select one caption, resulting in 591,435 image-text pairs, five times the total number of images.

**Visual Genome** (Krishna et al., 2017) Visual Genome stands as a meticulously labeled image dataset, enriched with detailed annotations for objects. Our employed training set features approximately 100K images, each accompanied by an average of 8 unique captions.

**CC3M** (Sharma et al., 2018) Also known as Conceptual Captions, the CC3M dataset contains approximately 3.3M image-text pairs. CC3M is collected from the web and carefully processed to achieve the cleanliness and informativeness of captions. Due to some broken image links, the final successfully downloaded dataset contains approximately 2.8M pairs.

**CC12M** (Changpinyo et al., 2021) The CC12M dataset is an extension of the CC3M (Sharma et al., 2018), hosting nearly 12M image-text pairs. A more relaxed collection pipeline was used for its creation. After downloading, our dataset comprises approximately 6M pairs.

**Laion2B** (Schuhmann et al., 2022) Laion-2B is a massive dataset populated with image data sourced from publicly accessible areas of the internet. Our successful download yielded a vast collection of around 1.3B images.

**Laion-COCO** (Christoph et al., 2022) Laion-COCO is a subset of the Laion-2B dataset which includes 600M image entries. These images have been captioned using the BLIP (Li et al., 2022) to generate high-quality descriptions mimicking the MS COCO style.

**Wukong** (Gu et al., 2022) Wukong is a 100M image-text pair in Chinese, where the images are filtered according to the image size, and the text is filtered according to its language, length, and frequency.

Table 6: Evaluation of bilingual multimodal chat models on UniMM-Bench in English.

| Model | OKVQA | AOKVQA | GQA | VQAv2 | AVG |
|---|---|---|---|---|---|
| mPLUG-Owl | 66.7 | 62.5 | 63.0 | 66.0 | 64.6 |
| VisualGLM | 57.6 | 62.8 | 58.1 | 63.9 | 60.6 |
| Ziya-Visual | 66.1 | 68.7 | 67.4 | 67.3 | 67.4 |
| Qwen-VL-Chat | **71.4** | **77.9** | 68.6 | **77.9** | **74.0** |
| VISCPM-Chat | 65.4 | 75.5 | **71.5** | 76.6 | 72.3 |

Table 7: Summary of multimodal chat models' architecture, parameters, and training data. Here, the size of mPlug-Owl's training data and the proportion of Chinese and English in the Ziya-Visual's training data are reported in their model cards.

| Model | Visual Module | LLM | Training Data |
|---|---|---|---|
| mPlug-Owl | ViT-L/14 (0.3B) | BLOOMZ-7B | - |
| VisualGLM | Q-Former (1.6B) | ChatGLM-6B | English: 300M; Chinese 30M |
| Ziya-Visual | Q-Former (1.1B) | Ziya-LLaMA-13B-v1 | 20M |
| Qwen-VL-Chat | ViT-bigG (1.9B) | Qwen-7B | English: 1.1B; Chinese: 300M |
| VISCPM-Chat | Muffin (0.7B) | CPM-Bee-10B | English: 140M; Chinese: 1M |

**Zero** (Xie et al., 2022) Sourced from a search engine, the Zero dataset comprises 20M images and corresponding textual descriptions, selected from a pool of 5B image-text pairs based on user click-through rate.

## B.2 INSTRUCTION TUNING DATASET

**LLaVA-Instruct-150K** (Liu et al., 2023a) LLaVA-Instruct-150K is a set of multimodal instruction-following data generated by GPT4. In its creation, image captions and corresponding bounding boxes are harnessed to encode images into textual sequences for the text-only GPT4 model. LLaVA-Instruct-150K divides the dataset into three types: Conversation, Detailed Description, and Complex Reasoning. For each type, manually designed examples are incorporated into prompts for in-context learning of GPT4.

**M³IT** (Li et al., 2023b) The M³IT dataset is a collection of large-scale multimodal instruction tuning datasets curated by leveraging downstream datasets covering diverse vision-language tasks, including captioning, reasoning, and visual question-answering. The datasets have been reformulated into a unified image-text schema. Certain constructed instances, derived from key datasets, are translated into a total of 100 languages. It is important to note that we do not proceed with additional translations for non-Chinese portions of the dataset. It's worth noting that we do not proceed with additional translations for non-Chinese portions of the dataset.

**UniMM-Chat** (Yu et al., 2023a) UniMM-Chat is a bespoke knowledge-intensive multimodal conversational dataset. UniMM-Chat utilizes images from the MSCOCO (Lin et al., 2014) and associated labeled datasets as the foundation for data generation. The related datasets include VQAv2 (Antol et al., 2015), Visual Dialogue (Das et al., 2017), OKVQA (Marino et al., 2019), and AOKVQA (Schwenk et al., 2022). The initial step involves merging the annotation information across different datasets for the same image, including pairs of questions and answers, dialogues, rationale information, and image captions. The varied text annotations provide a diversified view of the image content, enabling the comprehensive interpretation of the image by ChatGPT.

Table 8: Summary of text-to-image models' architecture and training data.

| Model | Visual Module | Text Encoder | Training Data |
|---|---|---|---|
| AltDiffusion | UNet | AltCLIP | - |
| TaiyiDiffusion | UNet | Taiyi-CLIP-RoBERTa | English: 0; Chinese: 20M |
| VISCPM-Paint | UNet | CPM-Bee | English: 1.3B; Chinese: 0 |

Table 9: Desription and examples of different categories in Chinese Drawbench

| Category | Description | Example |
|---|---|---|
| Relation | Ability to generate objects with specific interaction relationship between them | "挂在衣架上的一个帽子"
(A hat hanging on a hanger.) |
| Compositional | Ability to multiply kinds of objects | "一个盘子上放有黄色的梨和紫色的苹果"
(There are yellow pears and purple apples on a plate.) |
| Attribute | Ability to accurately generate objects with given attributes, such as size, color, and action. | "一件蓝色的天鹅绒晚礼服"
(A blue velvet evening gown.) |
| Counterintuitive | Ability to generate unusual scene against common sense | "穿着宇航服的李白开宇宙飞船"
(Li Bai flies a spaceship in a space suit.) |
| Rare Word | Ability to understand rare expression | 窗台上的一盆剑兰
(A gladiolus on the windowsill.) |
| Count | Ability to generate given numbers of objects | 三块涂有黄油的面包
(Three pieces of bread with butter.) |
| Long Input | Ability to understand long and complex input | 平静的湖面上，船夫划着船桨在湖面上划过，泛起涟漪，一只鸟飞过水面，抓起来一只鱼
(On the calm lake, the boatman rowed his paddle across the lake, rippling, a bird flew over the water and caught a fish.) |
| Chinese Culture | Ability to generate scene related to Chinese Culture | 海上生明月，天涯共此时
(As the bright moon shines over the sea,
From far away you share this moment with me.) |

Table 10: Details of image-text datasets used in VISCPM-Chat's pretraining, where "bil" means the mixed version of English and translated Chinese.

| Datasets | Size | Weight | Epoch |
|---|---|---|---|
| VISCPM-Chat | | | |
| COCO + Visual Genome | 626K | 12.50% | 27.60 |
| CC3M + CC12M | 8.4M | 25.00% | 16.40 |
| Laion-COCO | 390M | 62.50% | 0.35 |
| VISCPM-Chat+ | | | |
| COCO + VG | 626K | 12.50% | 15.33 |
| CC12M | 5.6M | 25.00% | 3.43 |
| Zero + Wukong | 20M | 12.50% | 0.48 |
| CC3M-bil | 5.6M | 12.50% | 1.72 |
| Laion-COCO-bil | 780M | 37.50% | 0.04 |

## B.3 CASE STUDY

Fig. 7 and Fig. 8 vividly depict VISCPM-Chat's capabilities, embodying wide-ranging global knowledge assimilation. As illustrated in the two case studies in Fig. 7, VISCPM-Chat can identify the Mona Lisa painting adapted in a surreal style and recognize a stained map of New York City, further interpreting its real-world meaning. Demonstrating a well-balanced capacity for Chinese-English multi-modal conversation and robust text recognition skills, as exemplified in Fig. 8, VISCPM-Chat can engage in fluent multimodal conversations on varying topics in English and effectively identify the words "Starbucks", "Avengers: Endgame", and its release date, "April 26th", from visual inputs.

## C DETAILS OF BASELINES

The details of existing Chinese-English bilingual multimodal chat models are summarized in Table 7 and introduced as follows:

- mPLUG-Owl (Ye et al., 2023): mPLUG-Owl consists of a vision encoder, a vision abstractor module, and BLOOMZ (Muennighoff et al., 2023) as language model backbone. It is trained on LAION-400M, COYO, CC3M, and MSCOCO.

Table 11: Performance of MPM on IGLUE benchmark. We compare MPM with mUNITER, xUNITER (Liu et al., 2021), UC$^2$ (Zhou et al., 2021) and M$^3$P (Ni et al., 2021).

| Model | xNLI | | | | xGQA | | | | | | |
|---|---|---|---|---|---|---|---|---|---|---|---|
| | ARB | SPA | FRA | RUS | BEN | DEU | IND | KOR | POR | RUS | CMN |
| mUNITER | 46.7 | 57.0 | 59.4 | 51.7 | 3.1 | 24.0 | 9.4 | 4.2 | 13.7 | 8.5 | 7.0 |
| xUNITER | 52.0 | 58.9 | 63.3 | 59.7 | 10.8 | 34.8 | 33.7 | 12.1 | 22.1 | 18.8 | 19.6 |
| UC$^2$ | **56.2** | 57.5 | 69.5 | 64.9 | **20.0** | 42.9 | 28.7 | 21.4 | 30.4 | 31.0 | 31.2 |
| M$^3$P | 55.2 | 58.9 | 56.4 | 62.5 | 18.6 | 33.4 | 32.5 | 25.1 | 31.4 | 27.5 | 28.7 |
| MPM | 55.8 | **75.9** | **78.1** | **75.0** | 6.6 | **51.0** | **37.1** | **36.9** | **49.8** | **47.9** | **46.1** |

- VisualGLM (Du et al., 2022): VisualGLM employs Q-Former as image encoder and ChatGLM-6B (Du et al., 2022) as language model backbone. VisualGLM-6B's pretraining incorporates 30M high-quality Chinese image-text pairs and 300M filtered English image-text pairs. In the fine-tuning phase, VisualGLM is trained on long VQA data.

- Ziya-Visual (Wang et al., 2022a): Ziya-Visual leverage Q-Former (Li et al., 2023a) as image encoder and Ziya-LLaMA-13B-v1 as language model backbone. They utilize 20M Chinese image-text pair, which is built by cleaning high-quality data from open source data, translating English datasets, and extracting coarse-grained information from captions using BLIP (Li et al., 2022) and Grounded SAM (Kirillov et al., 2023; Liu et al., 2023c).

- Qwen-VL-Chat (Bai et al., 2023): Qwen-VL-Chat utilizes a large ViT (Dosovitskiy et al., 2020) with 1.9B parameters initialized from Openclip's bigG (Ilharco et al., 2021) as image encoder and Qwen-7B as language model backbone. The image encoder and language model are connected by a one-layer cross-attention module augmented with positional encodings. Qwen-VL-Chat p

The details of existing Chinese-English bilingual text-to-image models are summarized in Table 8 and introduced as follows:

- AltDiffusion (Chen et al., 2023): AltDiffusion is a Chinese-English bilingual text-to-image model based on Stable Diffusion and bilingual vision-language encoder AltClip (Chen et al., 2023). The training data are collected form Laion (Schuhmann et al., 2021).

- TaiyiDiffusion (Wang et al., 2022a): TaiyiDiffusion is a Chinese text-to-image model that adapts a Chinese text encoder into Stable Diffusion. The visual part is frozen during training. The training datasets include 20M filtered Chinese image-text pairs.

# D DETAILS OF VISCPM-CHAT

## D.1 MULTIMODAL PRETRAINING

In the visual feature alignment pretraining, we train the visual encoder on a mix of image-text pair datasets, including CC3M (Sharma et al., 2018), CC12M (Changpinyo et al., 2021), COCO (Lin et al., 2014), Visual Genome (Krishna et al., 2017), and Laion-COCO (Christoph et al., 2022). We train VISCPM-Chat and VISCPM-Chat+ for 180K and 480K steps, respectively, with a batch size of 768 and a learning rate of 1e-5. The distribution of each dataset used in pretraining for VISCPM-Chat and VISCPM-Chat+ are shown in Table 10. Only the parameters of the visual module are optimized in this stage.

## D.2 INSTRUCTION TUNING

During the instruction tuning phase, the optimization process involves the visual module and the LLM. We train VISCPM-Chat on the mixed dataset from LLaVA-Instruct-150K, UniMM-Chat, and M$^3$IT, where LLaVA-Instruct-150K and UniMM-Chat are Chinese-English bilingual versions. The instruction tuning lasts for 80K steps with a batch size of 64. The training configuration of VISCPM-Chat+ is identical to VISCPM-Chat in this stage. In the inference phase, VISCPM-Chat uses beam search decoding with a beam size of 3.

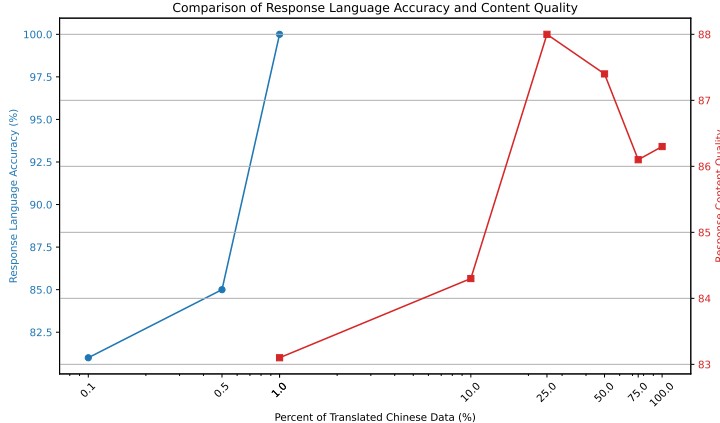

Figure 5: The trend of response language accuracy and content quality when using different percentages of translated Chinese data during instruction tuning.

### D.3 EVALUATION BENCHMARK

**LLaVA Test Set** (Liu et al., 2023a) LLaVA Test Set consists of 90 instances, each containing an image, a question, and an answer, which comprehensively evaluates the model's performance in conversation, detailed description, and complex reasoning. Following LLaVA (Liu et al., 2023a), we use GPT-4[3] to rate the model-generated answers and reference answers in a range of 1-10. We norm the average score of each instance to 1-100.

**UniMM-Bench** (Yu et al., 2023a) UniMM-Bench consists of 400 test instances, uniformly sampled from 4 commonly used VQA benchmarks, including OKVQA (Marino et al., 2019), AOKVQA (Schwenk et al., 2022), GQA (Hudson & Manning, 2019), and VQAv2 (Antol et al., 2015), whose annotations has undergone meticulous check. In the traditional evaluation of visual question answering, the metrics are to compute whether the model-generated answers are exactly matched to the reference answers. However, considering that LLMs typically generate responses with one complete sentence to answer the question, UniMM-Bench is built to comprehensively evaluate multimodal chat models' capabilities in the context of visual question-answering, including reasoning, commonsense, grounding, and world knowledge (Schwenk et al., 2022). Given the cost of using GPT-4 for evaluation, 400 test instances are sampled from reliable VQA benchmarks.

### D.4 THE TREND OF USING DIFFERENT NUMBER OF CHINESE TRANSLATED DATA

To investigate the influence of varying amounts of Chinese data on the model's ability to respond in the target language during the instruction tuning, surpassing the quasi-zero-shot transfer threshold, we systematically varied the percentages of translated Chinese data. Our analysis, depicted in Fig 5, elucidates the trends in both response language accuracy and content quality as a function of the Chinese data percentage utilized during instruction tuning. Remarkably, introducing a mere 0.1% of translated Chinese data suffices to achieve over 80% accuracy in response language accuracy, signifying the model's high sensitivity to even minimal language-specific data inputs. Elevating this percentage to 1% results in a scenario where all responses are rendered in Chinese. Regarding content quality, our findings suggest that a range between 25% to 50% of translated Chinese data represents an optimal balance, yielding high-quality content responses. We leave the exploration of determining a more precise optimal proportion of translated Chinese data for maximizing both response content quality in future work.

### D.5 ADDITIONAL RESULTS ON IGLUE

We evaluate the effectiveness of MPM in multilingual image understanding on two benchmarks of IGLUE: xVNLI (Bugliarello et al., 2022) and xGQA (Pfeiffer et al., 2022). We choose our model

---

[3]We use GPT-4 0314 version to evaluate the answers.

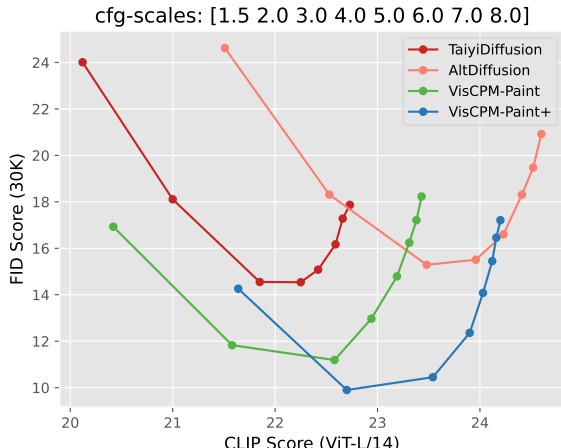

Figure 6: The curves of FID ↓ *vs* CLIP score ↑ with different classifier guidance scales.

pre-trained exclusively on English image-text pairs with LLaMA as the backbone language model, fine-tune it with IGLUE's English training set, and conduct a zero-shot evaluation following the official setting of IGLUE.

As Table 11 shows, the performance of MPM demonstrates a notable improvement over baseline models. Specifically, when using Chinese-English bilingual LLM CPM-Bee as the backbone language model, MPM attains a 56.05 score on xGQA for CMN (Chinese Mandarin). Based on the above results, we can observe that for languages where multilingual LLMs demonstrate proficiency, such as Spanish, French, Russian, and German, the multilingual capability can be effectively generalized to achieve strong multimodal capabilities in the target language.

# E    DETAILS OF VISCPM-PAINT

## E.1    TRAINING AND SAMPLING

VISCPM-Paint consists of a UNet, initialized from Stable Diffusion, and a bilingual LLM CPM-Bee (Zhang et al., 2021). We pretrain VISCPM-Paint on English image-text pairs dataset Laion-2B (Schuhmann et al., 2022) for 300k steps with batch size 4096. The 300K step consists of 200K-step training on 256×256 resolutions and 100K training 512×512 resolutions. We only optimize the cross-attention layer of UNet and the linear transblock between LLM and UNet. Linear transblock consists of a linear layer and a layer-norm layer, where the linear layer converts the dimension of LLM's hidden states into UNet's dimension. After pretraining on Laion-2B, we continue to train VISCPM-Paint+ on 20M filtered native Chinese image-text pairs and 136M translated image-text pairs in Laion-COCO (Christoph et al., 2022). We sample the generated image with DDPM scheduler (Ho et al., 2020) with 50 steps. For each prompt, we run three models to generate four images and select the one with the highest CLIP score evaluated by Chinese-CLIP (Yang et al., 2022).

## E.2    TRADE OFF OF FIDELITY AND ALIGNMENT

To visualize the trade-off between fidelity and alignment, Fig. 6 plots the curves of the FID against the CLIP score under different classifier guidance scales. The results indicate VISCPM-Paint and VISCPM-Paint+ deliver good overall performance regarding the balance between image quality and semantic alignment. While the CLIP score of AltDiffusion slightly outperforms that of VISCPM-Paint and VISCPM-Paint+, AltDiffusion's FID falls significantly short. The sub-optimal quality of generated images will affect the practical use, as shown in the human evaluation below.

### E.3    Human Evaluation

To better understand the model performance on text-to-image generation in Chinese, we create a comprehensive set of prompts in Chinese, named *Chinese Drawbench*. *Chinese Drawbench* contains 174 prompts and consists of 8 categories, including Relation, Compositional, Attribute, Counterintuitive, Rare Word, Count, Long Input, and Chinese Culture. The descriptions and examples of each category are shown in Table 9. During the construction of *Chinese Drawbench*, we asked 7 annotators to create 25 prompts per person conditioned on the given categories. To diversify the prompts, we also restrain the field of 20 prompts ranging from 9 common classes, such as animal, plants, food, space, etc, and leave five prompts with no restrain. After this, we double-check the prompts and polish them minor.

We invite 5 independent human evaluators to assess the performance of VISCPM-Paint, TaiyiDiffusion (Wang et al., 2022a), and AltDiffusion on *Chinese Drawbench*. For each instance, three random shuffled images and the input prompt are shown to the human evaluators. We ask them to select the best one among three images from 3 aspects: Overall, Alignment, and Fidelity. Alignment measures the consistency between the generated image and the input prompt. Fidelity measures the clarity, aesthetics appeal, and object realism of the image. Overall measures the comprehensive quality of generated images. Figure 10 provides detailed results of each category of *Chinese DrawBench*.

## F    Detail of Ablation Study

We introduce the detailed experimental configuration in the ablation study.

For image-to-text tasks shown in Table 5a, the native Chinese dataset consists of 20M image-text pairs filtered from Wukong (Gu et al., 2022) and Zero (Xie et al., 2022). For the "Pretrain" column in Table 5a, "Native Chinese" stands for the 20M native Chinese image-text pairs filtered from Wukong (Gu et al., 2022) and Zero (Xie et al., 2022). "English" corresponds to the dataset used in VISCPM-Chat, i.e., COCO, Visual Genome, CC3M, CC12M, and Laion-COCO. "Translated Chinese" consists of 390M Chinese image-text pairs whose captions are translated from English in Laion-COCO. In instruction tuning, i.e., supervised fine-tuning (SFT), we only utilize LLaVA and its translated Chinese version to reduce the computation time in the ablation study. Specifically, "Chinese" in the "SFT" column of Table 5a means translated Chinese LLaVA instruction tuning dataset. "Bilingual" means the mixed version of English and translated Chinese LLaVa.

For text-to-image tasks shown in Table 5b, "English" stands for Laion-2B. The other setting is the same as the image-to-text tasks introduced above.

## G    Detail of CLIP Score

We use Chinese-CLIP (Yang et al., 2022) to compute the CLIP score in Chinese image-text pairs. Chinese CLIP models have been used in several Chinese text-to-image models for data filtering and text encoding (Wang et al., 2022a; Liu et al., 2023b; Yang et al., 2023). We referred to Taiyidiffusion's criteria for data volume and filtering standards (including CLIP score) in training Chinese text-to-image models. We manually check the caption in the Wukong dataset. We observe that the overall quality of text content in Wukong is unsatisfactory. Here are some findings about low CLIP scores:

- Broken Images: Some image URLs are broken or redirected to the same default image for downloading.
- Unrelated captions: Some content of captions are totally unrelated to image content.
- Incomplete sentences: Some captions are truncated or missing punctuation.

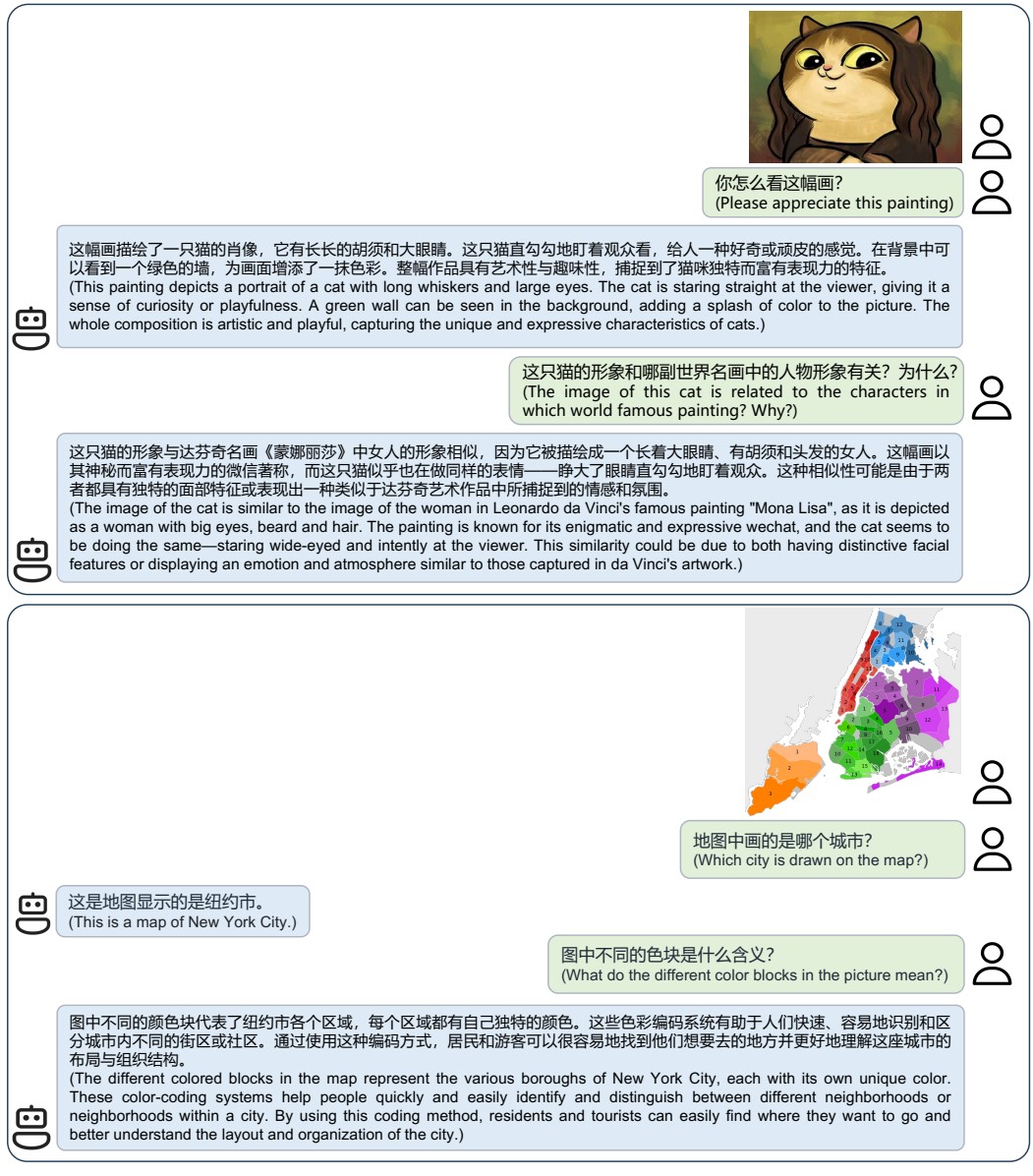

Figure 7: Multimodal conversation cases of VISCPM-Chat in Chinese. VISCPM-Chat can recognize artwork like the Mona Lisa, albeit adapted in a surrealistic style, and a detailed, stained map of New York City.

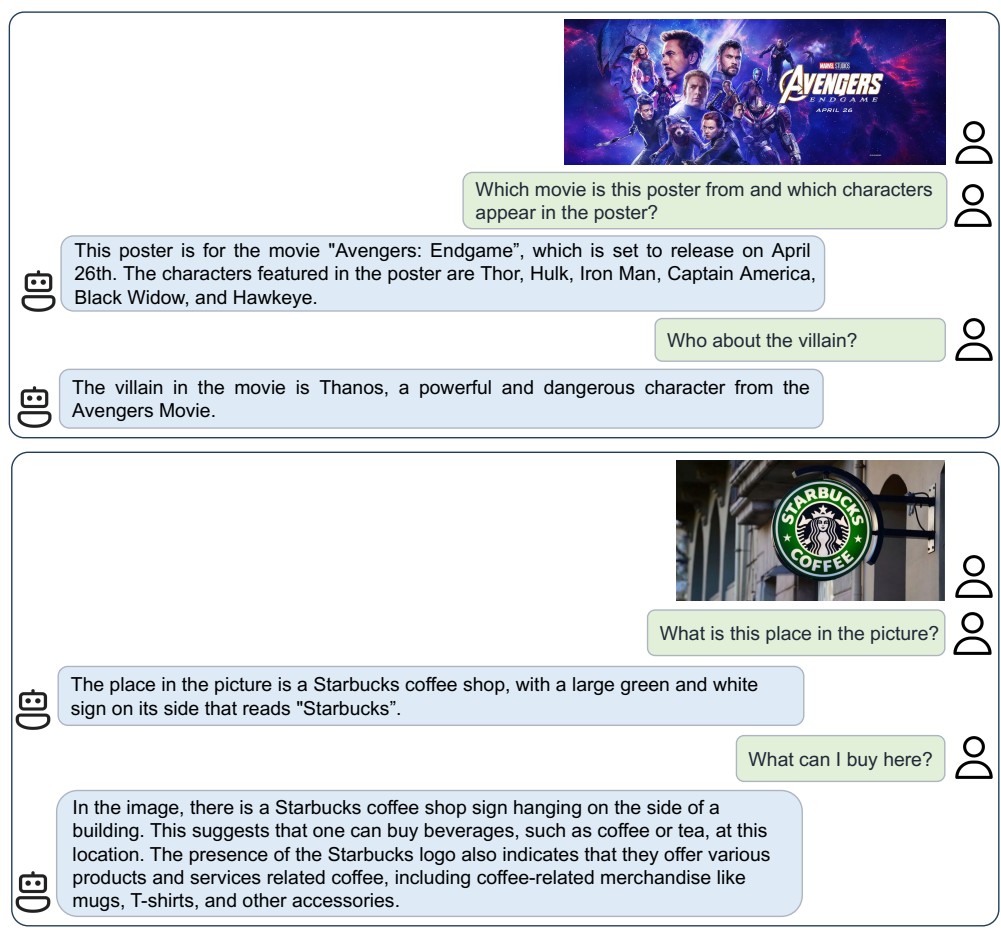

Figure 8: Multimodal conversation cases of VISCPM-Chat in English. VISCPM-Chat can conduct fluent multimodal conversations on different topics in English and recognize the words "Starbucks", "Avengers: Endgame" and its release date "April 26th" in the image.

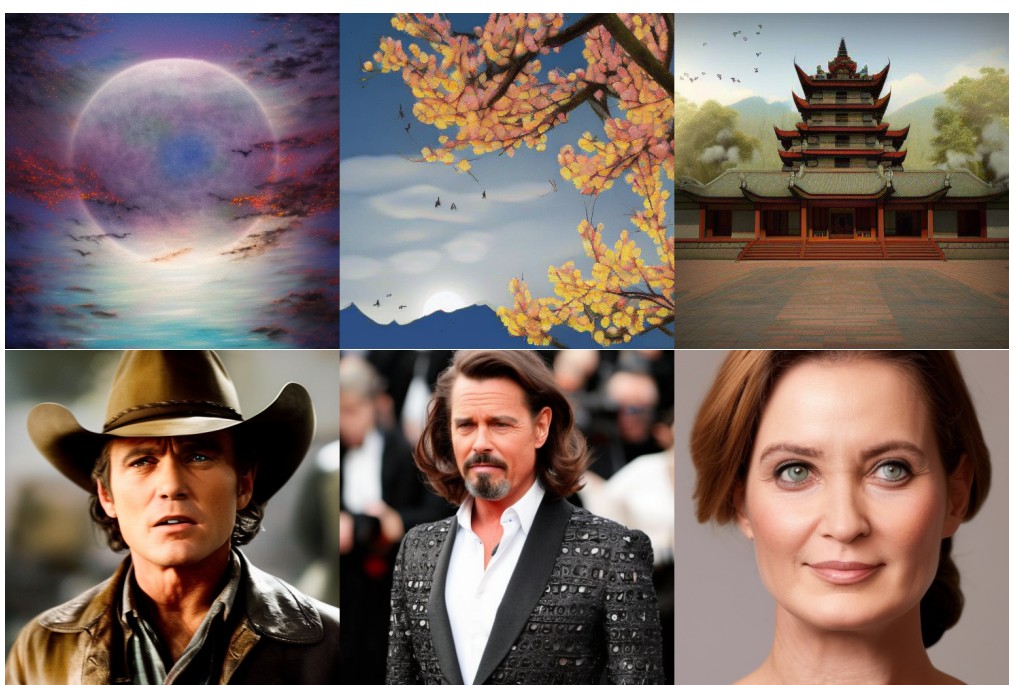

Figure 9: Generated images of VISCPM-Paint.

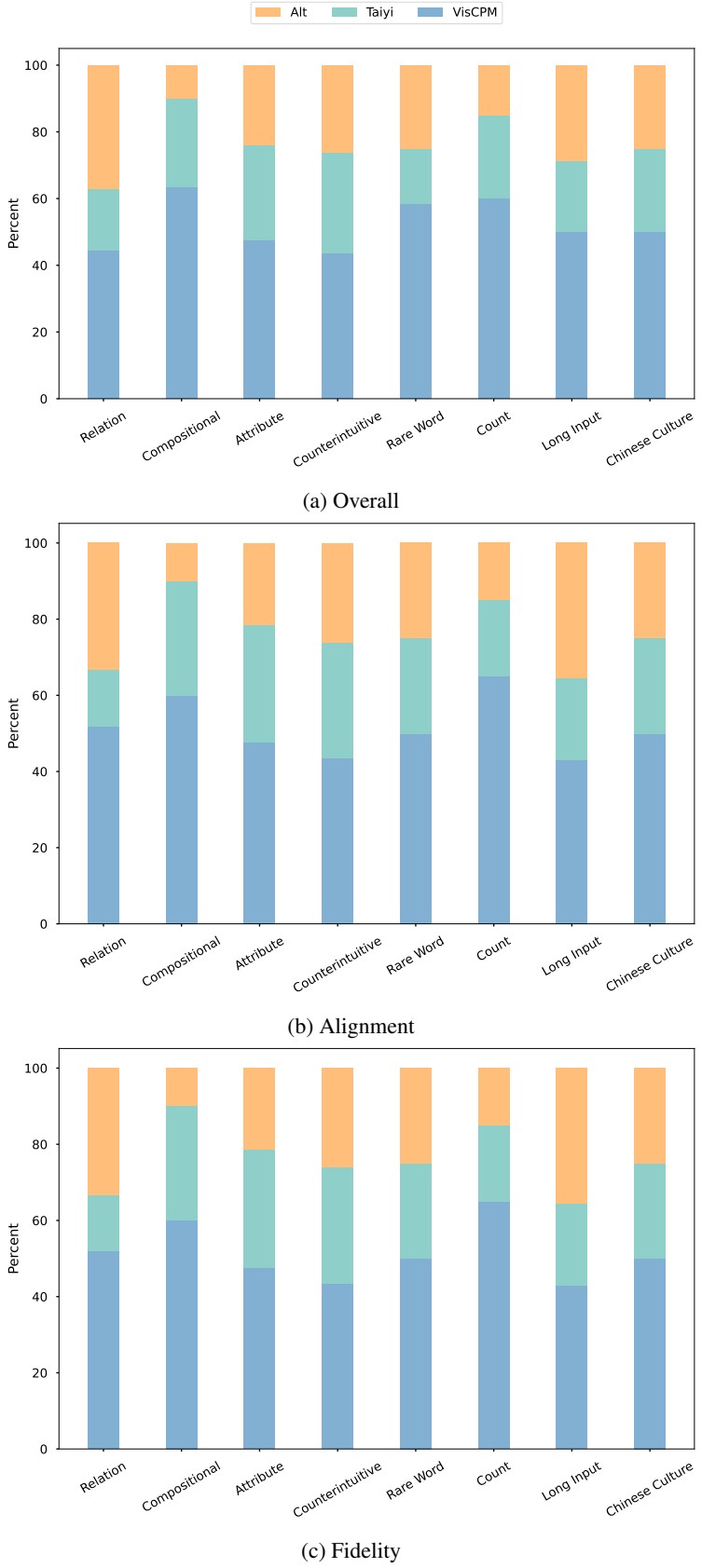

Figure 10: Human preference rate on Chinese Drawbench in different kinds of aspects.

