# OpenReview forum: "Large Multilingual Models Pivot Zero-Shot Multimodal Learning across Languages"
_ICLR.cc/2024/Conference — ICLR 2024 spotlight_

### Official Review · Reviewer_BZ1h · 2023-10-27

**Soundness:** 2 fair
**Presentation:** 3 good
**Contribution:** 3 good
**Rating:** 3
**Confidence:** 4

**Summary:**

This paper introduces MpM, a framework to train multilingual multimodal models for image-to-text and text-to-image generation (mostly English+Chinese). For image-to-text, the model (VisCPM-Chat) is typically pretrained on English image captions by freezing the LM, and then fine-tuned on instruction data translated into the target language(s). The model shows competitive performance when evaluated on machine-translated versions of LLaVA Test Set and UniMM-Bench as evaluated by an off-the-shelf model (GPT-4). For text-to-image, both the text model and the image decoder are frozen, and cross-attention layers are trained using English data. The resulting model (VisCPM-Paint) achieves a lower FID score than previous models on 30K samples from COCO Val obtained through machine translation.

**Strengths:**

1. The proposed framework, MpM, corroborates and extends the line of work on multilingual transfer through frozen language models, showing it to be more effective than previous models on the chosen tasks according to the evaluation measures adopted by the authors.
2. The plot in Figure 2 is interesting, showing how most of the Wukong text data might be poorly aligned with the corresponding images.
3. Human evaluation on Chinese Drawbench shows how the proposed model is often preferred to the baselines.
4. It is interesting to see how much English data helps during fine-tuning in the ablation study, perhaps because it bridges the gap between the style and language axes.

**Weaknesses:**

1. The paper claims to “generalize to other languages in a zero-shot manner.” This is not true, and very misleading. In fact, the chat model had to be fine-tuned on target language data, otherwise it would generate text in English. Calling the behavior “quasi-zero-shot” is also (i) different from the literature, and especially (ii) misleading: the model cannot generate text in the target language without supervision. Such claims, while valid for the text-to-image model, should be removed.
2. The paper claims to propose “an effective training paradigm for training large multimodal models in low-resource languages.” However, none of the languages tested were low-resource. There is a significant difference in the representations of the LM model between high- and low-resource languages, as well as in the quality of translations that an MT system can provide for those languages. None of these things are true for Chinese. Any claims about performance and applicability to low-resource languages need to be removed as they are never verified.
3. The paper also claims to be multilingual. Yet, most of the experiments are done in a bilingual setup (English+Chinese) or in a few, high-resource European languages. These languages share the same script, similar topology. Claims about multilinguality should be supported by a better selection of languages that increase diversity in scripts, topology, and resource availability.
4. My other major concerns are related to the experimental setup.
- (4a) The model is evaluated on unusual benchmarks for multilingual multimodal modeling. These datasets are obtained through machine translation of the corresponding English dataset. The authors claim to fix minor errors (only for Chinese, and not even for all the other 6 European languages), yet do not provide any indication of the quality of the resulting test data.
- (4b) The evaluation scores are based on GPT-4, which might not understand languages beyond English well. This is an hypothesis that might or might not hold true, but the fact that such concern exists, I believe it is enough to undermine the evaluation setup used by the authors.
- (4c) There exists multilingual multimodal benchmarks. For instance, IGLUE [1] tests for 4 multimodal tasks in 20 languages, with data *manually* collected by native speakers. MaRVL [2] and XM3600 [3] even include images sourced from the countries where a language is spoken. These benchmarks are adopted in the literature and completely missing (even as part of the related work) in this paper.
5. There is a lack of discussion of relevant related work. Models like {m,x}UNITER [2], CCLM [4] and TD-MML [5] are missing. The latter in particular shows how machine translation data can be very helpful during pretraining if filtering out poor translations, which provides a contrasting point to the one made by the authors in this paper. A discussion of these findings and how they differ is important to guide the community.

---

[1] Bugliarello et al. IGLUE: A Benchmark for Transfer Learning across Modalities, Tasks, and Languages. ICML’22.

[2] Liu et al. Visually Grounded Reasoning across Languages and Cultures. EMNLP’21.

[3] Thapliyal et al. A Massively Multilingual Multimodal Evaluation Dataset. EMNLP’22.

[4] Zeng et al. Cross-View Language Modeling: Towards Unified Cross-Lingual Cross-Modal Pre-training. ACL’23.

[5] Qiu et al. Multilingual Multimodal Learning with Machine Translated Text. EMNLP’22.

**Questions:**

1. The low CLIP scores in Wukong might be related to misunderstanding of Chinese captions from CLIP. You say that you perform manual inspection, can you elaborate more on your findings and their relation to CLIP scores?
2.  It would be useful for the reader to have more information about the baselines, so that one can easily understand what changes among models when comparing their results.

---

> ### Author Response · Authors · 2023-11-21
> **Response to Reviewer BZ1h**
>
> Thanks for your time and constructive reviews. Following, we present the response to address your concerns and questions. If you feel our responses effectively address your concerns, please kindly reconsider your evaluation.
>
> **Q1: Claim about 'quasi-zero-shot' Term**
> In MLLM literature, to the best of our knowledge, there has been no report on the phenomenon where a model exclusively pre-trained on English multimodal data, can provide accurate responses in English for instructions in other languages. In this context, the foundational MLLM capability has been successfully generalized from English to the target language, requiring merely a switch in language to fully realize this capability. On the one hand, this demonstrates a remarkable strength: the model's ability to comprehend and respond to Chinese multimodal instructions without being exposed to any Chinese multimodal data. On the other hand, such capability is imperfect as the model cannot respond in Chinese, whereas only very little instructing tuning data is needed to switch the language. Therefore, we use the term “quasi-zero-shot” to describe this subtle capability.
>
> We appreciate your valuable feedback and have refined the claim to a more precise expression that "generalize to other languages in a (quasi-)zero-shot manner", based on the quasi-zero-shot performance on image-to-text generation and zero-shot performance on text-to-image generation.
>
> *****
>
> **Q2: Language Choice**
> - The claim about low-resource language. In this paper, we refer to the low-resource language under a multimodal scenario. Please refer to the general response for a more detailed discussion. We have revised our paper to avoid misunderstanding with the term ‘low-resource’ language.
> - Language Diversity. Thanks for your advice! We have expanded our evaluation on a larger range of languages based on the IGLUE benchmark, which covers more diverse languages and obtains good performance, such as Russian in Cyrillic script with a 47.68 score on xGQA. We leave a wider range of language models in future work.
>
> *****
>
> **Q3: Experimental Setup**
>   - **Quality of test data**
>   For the Chinese test data, we manually check the data quality. This operation has been adopted in previously well-recognized benchmarks such as IGLUE to ensure the test data quality.
>   For other languages, note that directly using machine translation to translate the test benchmark without manual checking is also widely adopted in the literature [1][2][3]. Nonetheless, we agree that evaluation on manually validated benchmarks can better support the findings. During the response period, we additionally evaluate the performance of these languages on manually validated benchmark IGLUE.
>   - **Using GPT-4 as evaluator**
>     - **Feasibility**: GPT-4 has been widely accepted as an evaluator for AI chatbots, as also discussed in AlpacaEval [4], MTBench [5], and LLaVA Bench [6]. For example, the AlpacaEval benchmark is an authoritative benchmark for evaluating LLMs in following open-ended instructions. It uses GPT-4 as an automatic evaluator and shows a 0.94 Pearson correlation with human judgment. Our focus is to evaluate the performance of models in open-ended multimodal chats. Compared with traditional string-match-based metrics, GPT-4 evaluation is found to be more robust against the high diversity of natural language.
>     - **Multilingual Capability**: As for the multilingual capability, GPT-4 presents its strong multilingual capability on MMLU across languages in its report, which shows only a slight drop in the language we chose compared with English. We note that previous works [7][8] in the literature also use GPT-4 (ChatGPT) to evaluate multilingual tasks. For example, GEMBA [7] is a machine translation metric based on GPT-4 evaluation. We believe these results from the literature show the soundness of using GPT-4 to evaluate mainstream European languages.
>   - **Performance on IGLUE benchmark**
>   Thanks for your valuable advice. We perform an additional evaluation on the IGLUE. Please refer to the general response for the results.
>
> [1] Ding, Ming, et al. Cogview: Mastering text-to-image generation via transformers. NeurIPS 2021.
>
> [2] Ye, Fulong, et al. AltDiffusion: A Multilingual Text-to-Image Diffusion Model. Arxiv 2023.
>
> [3] Žagar, Aleš, and Marko Robnik-Šikonja. Slovene SuperGLUE benchmark: translation and evaluation. LREC 2022.
>
> [4] AlpacaEval: An automatic evaluator of instruction-following models. https://github.com/tatsu-lab/alpaca_eval.
>
> [5] Zheng, Lianmin, et al. Judging LLM-as-a-judge with MT-Bench and Chatbot Arena. NeurIPS 2023.
>
> [6] Liu, Haotian, et al. Visual instruction tuning. NeurIPS 2023.
>
> [7] Kocmi, Tom, and Christian Federmann. Large language models are state-of-the-art evaluators of translation quality. EAMT 2023.
>
> [8] Lai, Viet Dac, et al. ChatGPT Beyond English: Towards a Comprehensive Evaluation of Large Language Models in Multilingual Learning. Arxiv 2023.

---

> > ### Author Response · Authors · 2023-11-21
> > **Response to Reviewer BZ1h Part II**
> >
> > **Q4: Missing related work**
> > Thanks for your suggestions. We will incorporate the missing related work and benchmark in the final version.
> >
> > *****
> >
> > **Q5: Inconsistency with the previous conclusion**
> > In our work, we use a powerful multimodal LLM (10B) as the backbone language model, which already possesses strong cross-lingual generalization capability. In this case, only a small number of data in the target language is needed during SFT periods to switch on the target language in form. In this context, introducing more translation data can only bring a little marginal improvement.
> > In contrast, TD-MML [1] employs translation data in the context of a traditional language understanding model with relatively limited capacity and generalization capability. For such models, introducing more translation data can lead to more substantial improvements.
> > Thanks for the suggestion. We will incorporate this discussion into our final version.
> >
> > *****
> >
> > **Q6: Findings of manual check about the CLIP score**
> > We use Chinese-CLIP [2] to compute the CLIP score in Chinese image-text pairs. Chinese CLIP models have been used in several Chinese text-to-image models for data filtering and text encoding [3][4][5].
> > We manually check the caption in the Wukong dataset. We observe that the overall quality of text content in Wukong is unsatisfactory. Here are some findings about low CLIP scores:
> > - Broken Images: Some image URLs are broken or redirected to the same default image for downloading.
> > - Unrelated captions: Some content of captions are totally unrelated to image content.
> > - Incomplete sentences: Some captions are truncated or missing punctuation.
> >
> > *****
> >
> > **Q7: More information about the baselines**
> > Thanks for your reminder. Due to space limitations, we leave the detailed introduction of baseline models in the Appendix. We will incorporate more crucial information about the baselines in the body during revision.
> >
> > [1] Qiu, Chen, et al. Multilingual Multimodal Learning with Machine Translated Text. EMNLP 2022.
> >
> > [2] Yang, An, et al. Chinese CLIP: Contrastive Vision-language Pretraining in Chinese. Arxiv 2022.
> >
> > [3] Zhang, Jiaxing, et al. Fengshenbang 1.0: Being the Foundation of Chinese Cognitive Intelligence. Arxiv 2022.
> >
> > [4] Yang, Yukang, et al. GlyphControl: Glyph Conditional Controllable Visual Text Generation. NeurIPS 2023.
> >
> > [5] Liu, Shanyuan, Dawei Leng, and Yuhui Yin. Bridge Diffusion Model: Bridge non-English Language-Native Text-to-Image Diffusion Model with English Communities. Arxiv 2023.

---

> ### Comment · Reviewer_BZ1h · 2023-11-23
>
> Thank you for the detailed response, and for running additional experiments (and sorry for my late reply)!
>
> I'm considering increasing my score according to your response. Yet, I have three large concerns still:
>
> **General response:** I do not agree that 100M image--text pairs are low-resource data. Take Swahili as an example, while there are 10s of GB of text data [1], can you find 100M image--text pairs for it?
>
> **Q1:** given that you fine-tune the model on multimodal instructions in the target language, I would strongly prefer not to have any "zero-shot" string in your adaptation method. I still find "quasi-zero-shot" misleading.
>
> **Q3:** The languages that you selected for IGLUE evaluations are mostly high-resource. Would it be possible to evaluate on MaRVL or WIT?
>
> ---
>
> [1] Conneau et al. Unsupervised Cross-lingual Representation Learning at Scale. ACL 2020.

---

### Official Review · Reviewer_T6ab · 2023-10-31

**Soundness:** 3 good
**Presentation:** 4 excellent
**Contribution:** 3 good
**Rating:** 8
**Confidence:** 4

**Summary:**

Text-to-image and image-to-text generation efforts have primarily been focussed on English only due to lack of large-scale, high quality data. This paper proposes MPM, an effective training paradigm for training large multimodal models in low-resource languages. They show that this technique enables competitive zero-shot performance on Chinese language as compared to models trained on the language data. This leverages the Bilingual Dual-coding Theory which states that visual semantics are largely language agnostic. MPM divides the non-English multimodal learning into two consecutive stages: multilingual alignment and multimodal alignment. The former focuses on building a multilingual model by using a pretrained multilingual large language model, while the latter culminates in a multimodal model spanning multiple languages on top of the multilingual model.

**Strengths:**

- This paper shows a simple yet effective technique of using a mulitlingual LLM as a pivot to transfer multilingual image-text alignment across different languages.
- The model is trained on English data only but still manages to perform better on Chinese tasks than models trained on Chinese data. This is highly encouraging in making the state-of-the-art techniques on image-to-text and text-to-image generation available in other languages.

**Weaknesses:**

None.

**Questions:**

None.

---

> ### Author Response · Authors · 2023-11-21
> **Response to Reviewer T6ab**
>
> Thanks for your recognition of our work!

---

### Official Review · Reviewer_mPhK · 2023-10-31

**Soundness:** 3 good
**Presentation:** 2 fair
**Contribution:** 3 good
**Rating:** 6
**Confidence:** 4

**Summary:**

This paper studies a simple idea to use multilingual language models as the pivot to connect different languages for multimodal applications. The idea is based on the assumption that visual semantics are language agnostic, thus aligning a visual modal to a pre-trained (and sometimes frozen) multlilingual language model could support multilingual applications even without aligned multimodal data.

**Strengths:**

1. This paper is an engineering style paper, with great qualitative examples. The idea itself is simple and intuitive (but however the execution and writing is a bit complicated, see the next section).
2. This paper promised to open-source the code and model weights. This is going to have a positive impact on the community thanks to the open sourced model weights.

**Weaknesses:**

1. The contribution of the MPM training paradigm is unclear. Taking the image-to-text generation as an example, in the multimodal pre-training step the proposal is to freeze multilingual LLM and only tune the visual module; in the instruction tuning step the proposal is to finetune everything on the datasets from pivot language and the target language. The multimodal pre-training step is quite straightforward as it aligns a trainable vision model to a pre-trained and frozen LLM; the instruction tuning step is a standard finetuning setup. If the contribution of this paper is to use a pre-trained multilingual LLM to support multilingual downstream tasks, then much more in-depth analysis should be done to justify the point. Examples include understanding the MPM design choices, how freezing LLM affects results, how translated pairs affect instruction tuning results, how many translated pairs are enough and what the trend looks like.
2. Most of the tables present absolute comparisons, showing how great the proposed method is. However that might be less interesting from a reader's perspective without proper baselines and ablation tests as discussed in the above section (due to different dataset mixtures and tuning tricks used). It's high recommended to take a few tables, and perform multiple in-depth analysis and comparisons with controlled study, to make sure there are enough take aways and confidence in the current MPM design choices.

**Questions:**

See the above sections.

---

> ### Author Response · Authors · 2023-11-21
> **Response to Reviewer mPhK**
>
> We thank the reviewer for the valuable suggestion and acknowledgment. Following, we present the response to address your concerns and questions.
>
> **Q1: Contribution of MPM training paradigm.**
> We clarify the contribution of MPM in the general response. Please refer to the general response.
>
> *****
>
> **Q2: Analysis of MPM effectiveness**
> To realize strong multimodal capability across different languages, data strategy is one of the most important choices, considering the highly imbalanced data resource situation in different languages. Therefore, we focus on this perspective, while answers to some advised questions can be found in the paper.
> - **How freezing LLM affects results**: We discuss the effect of freezing LLM in Sec. 3.2. Freezing LLM can speed up training and avoid the negative influence of the low-quality text in the multimodal pre-training dataset.
> - **How translated pairs affect instruction tuning results**: The effect of translated pairs is also discussed in Sec. 3.2. We found that translated pairs of instructing tuning datasets can switch the model to respond in the target language.
> - **How many translated pairs are enough and what the trend looks like**: Our experiments show that merely 150K translated pairs during the SFT period are enough to switch on capabilities in the target language. Investigating the tight bounds and trends is a good idea that can lead to deeper understanding and stronger conclusions. Due to the time limit during the response period, we will incorporate a more detailed investigation in the final version.
>
> *****
>
> **Q3: In-depth analysis and comparisons with controlled study**
> We highly agree that analysis in controlled comparison is valuable for readers. For the main focus, we conducted a comprehensive ablation study with controlled comparison to analyze the effect of data from different sources (Table 5 and discussion in Sec. 5.3), which yields the following findings:
> - We ablate the languages of the pretraining dataset. The results show that relying solely on a native Chinese dataset is insufficient for achieving good performance for both image-to-text and text-to-image tasks.
> - We ablate the languages of the instruction tuning dataset. The results show that English data also plays a crucial role in improving the chat capability of the model in Chinese during the instruction tuning stage.
> - We ablate the filter process to the native Chinese dataset. The results show that the filtering process applied to the native dataset is essential for the model performance in Chinese.
> - We ablate the translated dataset in the pretraining stage. The results show that incorporating the native Chinese dataset and the translated Chinese dataset during the pre-training stage yields a marginal improvement in the model performance.
>
> We believe the analysis above offers a helpful know-how handbook for building strong non-English MLLMs. We will explore more data and training strategies as kindly suggested in our future work.

---

> > ### Comment · Reviewer_mPhK · 2023-12-03
> >
> > Thank you for the response. The authors have answered my questions, though more results for "Q2: Analysis of MPM effectiveness" could make the submission stronger. Thus I would like to keep my original rating score 6.

---

### Official Review · Reviewer_scbv · 2023-11-01

**Soundness:** 2 fair
**Presentation:** 3 good
**Contribution:** 3 good
**Rating:** 5
**Confidence:** 4

**Summary:**

The paper presents a method for training a multilingual and multimodal model without needing to rely on large amounts of data in all of the target languages. Instead, the method relies on multilingual and multimodal pivoting, using a high-resource language to provide the target distributions for the lower-resource language. The method is realized in an image-text and text-to-image model named VisCPM-Chat/Paint, respectively. The model is trained by aligning the low-resource Chinese data against the high-resource English data using the CPM-Bee bilingual Zho/Eng language model; the method is also shown to generalize to LLaMA.

**Strengths:**

S1: Conceptually simple approach to learning a multilingual and multimodal model. Makes good use of the vast amount of English language resources.

S2: Extensive experiments with relevant benchmark datasets.

S3: The approach is shown to generalize to more than one language model (LLaMA and CPM-Bee)

**Weaknesses:**

W1: There are some inconsistencies in the argumentation used throughout the paper. For example, in Section 4.1, the authors argue that translating a large-scale pretraining dataset will consume substantial computing resources, which is a fair argument. However, in Section 4.2, the authors describe that they train a version of their model called VisCPM-Chat+ using 1
36 million translated examples from LAION-COCO. The authors could improve their argumentation if they clarify whether it
is or is not challenging to create such translated examples. Bear in mind that other researchers have already translated
pretraining datasets, e.g. Thapliyal+ EMNLP 2022.

W2: The motivation for this paper is to create multimodal models for low-resourced languages via the proposed pivoting method but the main languages used in the experiments are Chinese and English, neither of which could ever be described as low-resource languages. The remaining languages used for the experiments in Section 5.1.3 are also hardly low-resource: German, French, Spanish, Italian, and Portuguese. Using these languages and claiming that the method applies to low-resource settings affects the credibility of the conclusions that one can draw for actual low-resource languages. See, for example, Joshi+ ACL 2020 for a discussion on low-resource languages.

Joshi, Pratik, Sebastin Santy, Amar Budhiraja, Kalika Bali, and Monojit Choudhury. "The State and Fate of Linguistic Diversity and Inclusion in the NLP World." In Proceedings of the 58th Annual Meeting of the Association for Computational Linguistics, pp. 6282-6293. 2020.

Thapliyal, A. V., Tuset, J. P., Chen, X., & Soricut, R. (2022, December). Crossmodal-3600: A Massively Multilingual Multimodal Evaluation Dataset. In Proceedings of the 2022 Conference on Empirical Methods in Natural Language Processing (pp. 715-729).

**Questions:**

1. Why do you think Chinese can be considered a low-resource language? See W2 for more details on my concerns.
2. In Section 5.1, why is it methodologically sound to use GPT-4 to evaluate model output?
3. In Figure 2, why is a CLIPscore threshold of 0.18 used to define high-quality? How was this determined?
4. In Section 4.2, what was the dataset of 100M image-text pairs used to align with the frozen LLM for 180K steps?
5. Did you also use M2M-100 to translate the 136M examples in the COCO-LAION dataset?
6. Which version of M2M-100 did you use for translation?
7. In Section 5.2.2, you describe an FID of 9.5 or 9.9 "comparable" to Stable Diffusion FID is 8.6. Is this a reasonable claim?
8. How does your model perform on a larger set of languages in the multilingual multimodal benchmark IGLUE by Bugliarello+ ICML 2022?

Bugliarello, E., Liu, F., Pfeiffer, J., Reddy, S., Elliott, D., Ponti, E. M., & Vulić, I. (2022, June). IGLUE: A benchmark for transfer learning across modalities, tasks, and languages. In International Conference on Machine Learning (pp. 2370-2392). PMLR.

---

> ### Author Response · Authors · 2023-11-21
> **Response to Reviewer scbv**
>
> We are grateful for your time and the careful review. We present the response to address your concerns and questions. Please kindly reconsider your evaluation if you feel our responses effectively address your concerns.
>
> **Q1: Why translate the dataset?**
> It is certainly true that the translation of the 136M examples did consume substantial computational resources. In our experiments, we employed the bilingual LLM, CPM-Bee 10B, for translation. The translation process consumed 4,533 GPU hours on NVIDIA A100 80G for 136 million examples. We would like to clarify that the goal of this operation is to empirically validate the hypothesis of MPM in a controlled setting, verifying whether translated datasets are necessary for training MLLMs. Our experimental results show that marginal improvement can be achieved even with high-quality large-scale datasets translated by strong bilingual LLMs. Considering the cost vs improvement, the conclusion is that translating pretraining data is unfavored for training non-English MLLMs. We hope the empirical results contribute to a better understanding of the role of data translation in training non-English MLLMs.
>
> *****
>
> **Q2: The meaning of low-resource languages**
> Thanks for your suggestion. Please refer to the general response for a more detailed discussion. In general, we focus on the low-resource nature of languages in the multimodal scenario. We have revised our paper to avoid confusion with the terminology "low-resource languages" in pure language resources.
>
> *****
>
> **Q3: Methodological soundness to use GPT-4 for evaluation**
> GPT-4 has been widely accepted as an evaluator for AI chatbots, as also discussed in AlpacaEval [1], MTBench [2], and LLaVA Bench [3]. For example, the AlpacaEval benchmark is an authoritative benchmark for evaluating LLMs in following open-ended instructions. It uses GPT-4 as an automatic evaluator and shows a 0.94 Pearson correlation with human judgment. Our focus is to evaluate the performance of models in open-ended multimodal chats. Compared with traditional string-match-based metrics, GPT-4 evaluation is found to be more robust against the high diversity of natural language. Therefore, we follow previous works and use GPT-4 to evaluate MLLM conversation performance.
>
> *****
>
> **Q4: CLIP score threshold**
> We referred to Taiyidiffusion's criteria for data volume and filtering standards (including CLIP score) in training Chinese text-to-image models. We conducted a thorough manual check for the correlation between image-text alignment and the CLIP score. Here are some specific findings about image-text pairs with low CLIP scores:
> - Broken Images: Some image URLs are broken or redirected to the generic default image for downloading.
> - Unrelated captions: Some captions are entirely irrelevant to the associated image content.
> - Incomplete sentences: Some captions are truncated or missing punctuation
>
> *****
>
> **Q5: Dataset composition**
> The 100M image-text pairs used to align with the frozen LLM consist of COCO, Visual Genome, CC3M, CC12M, and Laion-COCO. We introduce related training details in Appendix Sec. C.
>
> *****
>
> **Q6: Choice of Machine Translation System**
> We use the CPM-Bee 10B for all En-Zh translations due to its better performance. For other languages, we use M2M100 12B for translation in German, French, Spanish, Italian, and Portuguese.
>
> *****
>
> **Q7: Conclusion about FID performance**
> We draw a conclusion on the performance comparison between our model (9.5/9.9 FID) and Stable Diffusion (8.6 FID) referring to the literature. For instance, Unidiffuser[4], with an FID of 9.7, concluded in their paper that their model's performance on single text-to-image generation tasks is "on par" with custom diffusion models such as Stable Diffusion.
>
> *****
>
> **Q8: Performance on IGLUE**
> Thanks for your advice. Please refer to the general response for the performance of MPM on IGLUE.
>
> *****
>
> [1] AlpacaEval: An automatic evaluator of instruction-following models. https://github.com/tatsu-lab/alpaca_eval.
>
> [2] Zheng, Lianmin, et al. Judging LLM-as-a-judge with MT-Bench and Chatbot Arena. NeurIPS 2023.
>
> [3] Liu, Haotian, et al. Visual instruction tuning. NeurIPS 2023.
>
> [4] Bao, Fan, et al. One transformer fits all distributions in multi-modal diffusion at scale. ICML 2023

---

### Author Response · Authors · 2023-11-21
**General Response to All Reviewers**

Thanks for all of your time writing valuable and constructive comments. Your feedback assists us in enhancing the quality of our paper, and thus, we are committed to incorporating your suggestions in our revision process. Meanwhile, we feel encouraged that the reviewers find our method simple and effective (Reviewer mPhK, T6ab, and BZ1h), and recognize the generalization of MPM across languages (Reviewer scbv and T6ab). Your support means a lot to us!
At this juncture, we would like to re-emphasize the significance of this work, the meaning of ‘low-resource language’ in this paper, and present the effectiveness of MPM on the IGLUE benchmark.

*****

**Contribution of MPM**
Multimodal large language models (MLLMs) have become one of the worldwide focus points in both the NLP and CV communities. There is a growing need to develop such models for non-English languages. Conventionally, this involves extensive pretraining on native multimodal data in the target language, which is exactly the training method of baseline models in our paper. However, due to the scarcity and low-quality of multimodal data in non-English languages, it is very challenging to build a powerful large multimodal model only based on native data.
To address the challenge, MPM leverages the power of large multilingual models and the abundance of English multimodal data. Unlike conventional methods, we exclusively utilize English multimodal data to pretrain non-English multimodal models. This approach, simple yet somewhat counterintuitive, has shown surprising effectiveness across a diverse range of languages. The most significant contribution of our work comes from this paradigm shift, which is central to MPM's methodology. We believe it provides a scalable and generic solution that can efficiently harvest the rapid advances in multilingual LLM and vision communities, and would bring inspiration to developing MLLMs across various languages and cultures.

*****

**The meaning of 'low-resource languages'**
In this paper, we discuss data resources in the context of multimodal learning, where low-resource languages mean those with limited multimodal data. In fact, apart from English, where billion-scale multimodal pretraining datasets are available, all other non-English languages lack sufficient data for independent large-scale MLLM training. Taking Chinese as an example, as analyzed in Sec. 4.1, the largest Chinese multimodal dataset, Wukong, only contains 100M pairs, most of which are of very low quality. Therefore, we call these languages as low-resource languages lacking multimodal data in this paper.
We appreciate the suggestions from Reviewer scbv and BZ1h and notice that it may lead to some confusion with the existing terminology 'low-resource languages' in pure language resources. We have revised our paper to make it clear for understanding.

---

> ### Author Response · Authors · 2023-11-21
> **General Response Part II**
>
> **Performance of MPM on IGLUE**
> We evaluate the effectiveness of MPM in multilingual image understanding on two benchmarks of IGLUE: xVNLI and xGQA. We choose our model pre-trained exclusively on English image-text pairs, fine-tune it with IGLUE's English training set, and conduct a zero-shot evaluation following the official setting of IGLUE. We report the performance of MPM in each language:
>
> **xVNLI**:
> | **Model** | *ARB* | *SPA* | *FRA* | *RUS* |
> |-----------|---------|---------|---------|---------|
> | mUNITER   | 46.73   | 56.96   | 59.36   | 51.72   |
> | xUNITER   | 51.98   | 58.94   | 63.32   | 59.71   |
> | UC2       | **56.19**   | 57.47   | 69.67   | 64.86   |
> | M3P       | 55.24   | 58.85   | 56.36   | 62.54   |
> | MPM       | 55.84   | **75.86**   | **78.09**   | **75.00**      |
>
>
> **xGQA**:
> | **Model** | *BEN* | *DEU* | *IND* | *KOR* | *POR* | *RUS* | *CMN*       |
> |-----------|---------|---------|---------|---------|---------|---------|---------------|
> | mUNITER   | 3.06    | 23.95   | 9.36    | 4.21    | 13.67   | 8.49    | 7.03          |
> | xUNITER   | 10.80    | 34.83   | 33.73   | 12.12   | 22.13   | 18.84   | 19.55         |
> | UC2       | **19.99**   | 42.85   | 28.67   | 21.36   | 30.42   | 31.00      | 31.16         |
> | M3P       | 18.64   | 33.42   | 32.48   | 25.11   | 31.40    | 27.50    | 28.65         |
> | MPM       | 6.60     | **51.04**   | **37.08**   | **36.85**   | **49.77**   | **47.68**   | **46.06 (56.05)** |
>
> The performance of MPM demonstrates a notable improvement over baseline models. Specifically, the 56.05 xGQA score for CMN (Chinese Mandarin) is attained using CPM-Bee as the backbone language model, while  LLaMA is utilized for other languages. Based on the above results, we can observe that for languages where multilingual LLMs demonstrate proficiency, such as Spanish, French, Russian, and German, the multilingual capability can be effectively generalized to achieve strong multimodal capabilities in the target language.
>
> *****
>
> In response to the reviews’ valuable feedback, we've carefully revised our manuscript and conducted additional experiments and evaluations. We hope our updates address your concerns. We kindly request a re-evaluation of our work, and look forward to further acknowledgment.
>
> Best regards.

---

### Meta-Review · Area_Chair_x55m · 2023-12-04

**Metareview:**

This paper aims to build multilingual multimodal LLMs for non-English languages. The authors reveal that multilingual LLMs can play a pivotal role in generalizing multimodal capabilities across languages, leading to strong results without language-specific multimodal pretraining. During the discussion phase, the authors properly responded and addressed the issues from the reviewers (i.e., clarifying the terms "low-resource languages" and "zero-shot", and supplementing experiment results on more languages). Reviewers BZ1h mentioned considering increasing the score at the discussion phase, under the condition that the terms are modified. Through the final response to chairs, the authors agreed to switch the terms to avoid potential misunderstanding. Therefore, I think this paper is entirely acceptable to the conference.


Strength of the paper:
1. This paper proposes a novel, sound and scalable approach to developing multilingual multimodal LLMs. The idea and findings about the ``quasi-zero-shot'' multimodal capability across languages are novel and interesting, and may inspire future research.
2. The experiments are solid with robust evaluation and strong results in both image-to-text and text-to-image generation. The strong results in non-English languages are particularly encouraging, considering that only English multimodal data is used in pretraining.
3. The codes and pretrained model weights will be open-sourced, which will have a positive impact on the community.

Weakness:
1. Most issues raised by reviewers are about the term "low-resource languages" used in the paper (refer to non-English languages in multimodal context), which is a bit different from how it's generally interpreted in community (languages lacking sufficient text data). Authors have modified the paper and addressed the issue during the discussion phase.
2. More detailed analysis of the design choices (e.g., how many translated pairs are enough to switch on the target language and what the trend looks like) could make the paper stronger.

**Justification For Why Not Higher Score:**

Some terms and expressions should be modified in the revision to avoid potential misunderstanding.
More detailed analysis of design choices could be made.

**Justification For Why Not Lower Score:**

The proposed framework is novel and easy-to-follow. The experiments are comprehensive with robust evaluation and strong results in both image-to-text and text-to-image generation. The framework and findings constitute a valuable contribution to developing MLLMs for non-English languages. Issues raised by the reviewers are addressed in the discussion.

---

### Decision · Program_Chairs · 2024-01-16

Accept (spotlight)